



**Predicting the amplitude and runup of the water waves induced by rotational**
**cliff collapse, considering fragmentation**
Hasnain Gardezi[a*], Talha Khan[c], Xingyue Li [a,b*], Taimur Mazhar Sheikh[a], Yu Huang [a,b], Zhiyi Chen [a]
[a] Department of Geotechnical Engineering, College of Civil Engineering, Tongji University, Shanghai
200092, China
[b] State Key Laboratory of Disaster Reduction in Civil Engineering, Tongji University, Shanghai 200092,
China
[c] School of Aerospace Engineering and Applied Mechanics, Tongji University, Shanghai 200092, China
*Corresponding authors: hasnain_haider@tongji.edu.cn (Hasnain Gardezi), xingyueli@tongji.edu.cn
(X.Y.Li)
**Abstract:**
Cliff collapse-induced water waves in small lakes and reservoirs retain their energy
due to short travel distance, and may cause significant damage to offshore infrastructure.
Previously, scientists have analyzed the waves induced by granular/block sliding down
the slope and hitting a water body, but none have studied the water waves induced by
rotational cliff collapse, fragmenting upon impact with the water surface. So, in this
study, we have experimentally and numerically analyzed the rotational cliff collapse
and energy transfer mechanism, determined the amplitude and runup of the induced
waves, and developed machine learning-based prediction models. Moreover, the effect
of the fragmentation of the cliff upon impact on the induced wave has also been
investigated. The results indicate that as the water depth decreases, the impact Froude
number and relative wave amplitude increase, wave velocity decreases, and the splash
becomes more elongated. A comparison between the wave induced by fragmented cliff
collapse and an equivalent amount of granular mass sliding from a 30° slope indicates
that the amplitude of the waves induced by granular mass is 42%, 35%, and 28% less
than that of fragmented cliff collapse. The wave amplitude induced by fragmented cliff
collapse indicates that the rotational motion of the cliff imparts a more sudden and
concentrated impact that allows an efficient energy transfer to water, resulting in higher
wave amplitudes. The results for the prediction model indicate that the amplitude and
runup model performed well both in the training and testing stages, with higher $R^2$





values. The developed model was validated by comparing the results with established
statistical indices and by performing sensitivity and parametric analysis, highlighting
that wave amplitude is greatly influenced by impact velocity, cliff height, and the
number of fragments, contributing approximately 90% to the wave amplitude. In
comparison, runup is greatly influenced by bank slope angle, impact velocity, cliff mass,
and height. The experimental results and developed prediction models can provide the
basis for understanding the rotational cliff collapse-induced waves and can help with
disaster mitigation and risk assessment by effectively predicting the wave amplitude
and runup.
Keywords: Rotational cliff collapse, wave amplitude, runup, cliff fragmentation,
prediction model.
**1. Introduction**
The phenomenon of cliff overturning is common along rivers and reservoirs (glacial
lakes, recreational lakes), and has been captured by various people around the globe.
The cliffs around these lakes are weathered due to climate change and wave action (Ró
and Cerkowniak, 2024; Young et al., 2021) and can no longer be supported by the parent
rock. When these initially intact, weathered cliffs fall into water, they usually fragment
upon impact with the water surface, and as a result, induce an impulse water wave.
Upon impact, the energy of gravitational mass is transferred to the water body, resulting
in a huge splash and a wave train, propagating away from the point of impact. In the
reservoirs and water channels located in mountainous regions, such as glacial lakes,
dams, and a river flowing through valleys, these waves do not travel a long distance
before reaching obstacles, opposite shores, or other infrastructure. As the waves retain
most of their energy, size, and strength, the impact can cause significant damage to the
population and infrastructure located along the banks of the reservoir. Historically,
extreme impulse wave heights have been observed induced by landslides in events of
1958 Lituya Bay, USA, which caused a wave height of 524 m (Boultbee et al., 2006;
Franco et al., 2020; Miller, 1960), 2007 Chehalis Lake, Canada, induced a wave of 38
m (Wang et al., 2015), 2015 Taan Fjord, USA, caused a wave of 193 m (Higman et al.,
2018), and 2014 Lake Askaja (Gylfadóttir et al., 2017).



Moreover, the highly energetic gravity waves are capable of overtopping the dam
wall, especially where the freeboard is just a few meters. The overtopping can result in
dam failure and can lead to catastrophic events, such as caused by 1963 Vajont rock
slide, in North Italy, where a 250 Mm$^3$ of rock mass slid into the dam reservoir and
induced a huge wave that ran to a height of 200 m at the opposite bank (Franci et al.,
2020; Heller and Ruffini, 2023; Ward, N. Steve and Day, Simon, 2011; Zhao et al.,
2016), the resultant wave overtopped the dam and destroyed an entire village
downstream. Similarly, in 2003 Qianjiangping landslide with a volume of 24 Mm$^3$, and
in 2008 Gongjiafang landslide with a volume of 0.38Mm$^3$ in Three Gorges dam
reservoir area induced a water wave that had an amplitude of 30 m and 32 m
respectively (Wang et al., 2021), Gongjiafang landslide induced wave ran up to a height
of 12.4 m on opposite bank (Huang et al., 2012).
These incidents highlight the need for predicting the subsequent energy transfer of
such cliff collapses for disaster mitigation. The wave amplitude and runup height are of
great importance. In contrast, the cases mentioned above are extreme; the small sliding,
toppling, and falling events in small lakes and reservoirs can induce a wave of
comparably small amplitude but capable of causing substantial damage to densely
populated areas along the shoreline. Particularly, in the case of glacial lakes,
recreational lakes, and lakes formed by previous landslides are prone to cause major
disasters as they are considerably smaller compared to dam reservoirs (Gardezi et al.,
2021). The phenomena of cliff overturning and falling are common around these lakes
and have been captured by various people around the globe. Fig. 1 (a, b, and c) indicates
a rotational (topple) cliff collapse in Furnas Lake, Brazil, on 8 January 2022, killing 10
people. As a result of the collapse, a huge splash and induced waves can be seen in Fig.
1 (c). Though scientists have analyzed the amplitude and runup of the waves induced
by sliding masses, the literature lacks in providing detailed information on the
formation and propagation of the wave induced by rotational cliff collapse. Moreover,
the literature also lacks in elaborating on the shape of the induced splash. Effect of cliff
fragmentation on the induced wave, as can be seen in Fig. 1 (a, b, and c), the falling
cliff was still intact and broke under its own weight upon impact with the water surface





and induced a huge splash. Furthermore, though there are numerous prediction models
available for the amplitude and runup of landslide-induced water waves, the prediction
models for water waves induced by rotational fall of cliffs considering fragmentation
are nonexistent.
Field data related to historical events is critical for disaster mitigation, but due to
their occurrence in remote areas, the unavailability of measuring devices makes it
difficult, leaving the physical modeling as the only source for understanding the wave
generation, and propagation phenomena (Bellotti and Romano, 2017; Grilli et al., 2017;
Takabatake et al., 2022; Wang et al., 2017a; Watts, 1998a). Previously, scientists have
performed both two, and three-dimensional physical modeling for landslide induced
water waves using either block slide (Heinrich, 1992; Heller and Spinneken, 2013;
Najafi-Jilani and Ataie-Ashtiani, 2008; Sælevik et al., 2009), (M. Di Risio et al., 2009;
Marcello Di Risio et al., 2009; Lindstrøm et al., 2014; Montagna et al., 2011; Panizzo
et al., 2005; Wang et al., 2016) or granular slide (Fritz et al., 2003a, 2003b; Lindstrøm,
2016; Miller et al., 2016; Zweifel et al., 2006),(Heller and Spinneken, 2015; Huang et
al., 2014; McFall and Fritz, 2016; Mohammed and Fritz, 2012; Romano et al., 2023).
But none have developed a physical model to quantify the amplitude and runup of the
waves induced by rotational cliff collapse, incorporating cliff fragmentation.
Along with the physical modeling, the wave generation and propagation
phenomena have also been analyzed using numerical modeling by using Eulerian and
Lagrangian methods, employing depth-averaged model, nonlinear shallow water,
Navier-Stokes model, or Boussinesq equation, for both two- and three-dimensional
modeling (Cecioni et al., 2011; Grilli et al., 2019; Heidarzadeh et al., 2020; Løvholt et
al., 2005; Ruffini et al., 2019; Watts et al., 2003; Whittaker et al., 2017; Yavari and
Ataie-Ashtiani, 2017). Moreover, numerous scientists have also used computational
fluid dynamics (CFD) methods to analyze wave phenomena, just like experimental
modeling, considering the sliding phase as solid material and water as the fluid phase
(Abadie et al., 2010; Chen et al., 2020; Clous and Abadie, 2019; Franci et al., 2020b;
Guan and Shi, 2023; Heller et al., 2016; Kim et al., 2020; Ma et al., 2015; Montagna et
al., 2011; Mulligan et al., 2020; Paris et al., 2021; Rauter et al., 2022; Romano et al.,



2020; Shi et al., 2016).
Furthermore, scientists have also developed empirical prediction models for
landslide-induced water waves by considering a combination of several parameters, i.e.,
geometric, geological, and kinematic characteristics of slides that contribute to wave
generation, as shown in Table 1. Scientists have M. M. Das and Wiegel (1972) proposed
that the velocity of the sliding material and water depth are the main components
affecting the amplitude of the waves. Watts (1998) Stated that the slope angle, length,
and mass of the slide are major factors influencing the amplitude of the wave. Fritz et
al. (2003) stated that the landslide mass thickness mainly drives the amplitude of the
induced wave. The empirical relations mentioned in Table 1 are mainly for the
landslide-induced water waves, not for cliff collapse-induced water waves. Since this
study is related to the wave induced by rotational cliff collapse, not the granular slide,
the contributing parameters should also be different. Here in this study, we have
considered seven parameters for developing a prediction model, i.e., water depth, fall
height of the cliff, number of fragments, runup slope angle, height of the cliff, and
impact velocity. In this study, we have incorporated a new parameter, i.e., the number
of fragments, as the induced waves from fragments better replicate actual geohazard
events.
Since the experimental and numerical models are expensive, laborious, time-
consuming, and require a lot of expertise, to overcome these problems, there is a need
for models that are quick and require less effort and cost. Consequently, the use of AI
and ML-based models is gaining fame in the field of engineering. Previous prediction
models for wave amplitude and runup employ simple regression analysis, which is
insufficient for complex problems involving multiple parameters, but recently scientists
have shifted towards more advanced ML models (Bujak et al., 2023; Cesario et al.,
2024; Li et al., 2024, 2023a; Romano et al., 2009; Tarwidi et al., 2023; Tian et al., 2025;
Wang et al., 2017b; Wiguna, 2022). Though scientists have used machine learning for
wave amplitude and runup prediction modeling induced by various types of gravity
flows, the prediction model for the waves induced by the rotational collapse of the cliff
involving fragmentation is nonexistent to the authors' knowledge. Here in this study,



we have developed prediction models for wave amplitude and runup using genetic
programming (GP).

GP-based models have recently gained traction for prediction, and multi-

expression programming (MEP) and genetic-expression programming (GEP) are the
most advanced, sophisticated, and widely used models. Both models are gene-type
programming models that form tree-like models. These models are similar to living
organisms, which can learn, adapt, and modify their composition, size, and shape
(Gardezi et al., 2024; Usama et al., 2023). MEP is a cutting-edge, advanced form of
GEP that adopts a demonstrative model for programming and uses linear chromosomes
to determine optimum population size, mutation probability, and evolutionary model.
Compared to other ML models, it can produce more precise results even when the
problem complexity is unknown (Usama et al., 2023).

In this study, we have experimentally and numerically analyzed the hydrodynamics

of the wave induced by rotational cliff collapse, and have also developed a prediction
model for wave amplitude and runup. The physical modeling was carried out by
developing a scaled water flume and a platform inducing rotational motion of the cliff.
A total of 162 experiments were carried out by varying the seven control parameters,
i.e., water depth, fall height, cliff mass, impact velocity, cliff height, runup slope angle,
and number of fragments; for the sake of accuracy, each experiment was conducted
twice, for consistency, thus making it 81 experiments. The parameters were selected to
comprehensively elaborate on the distinct phases of rotational cliff collapse and induced
waves. Water depth and runup slope angle provide the basis for wave propagation and
runup. Whereas, the cliff collapse dynamics are explained by cliff mass (which governs
the energy input), height of the cliff (defines the initial potential energy), and fall height
(determines the transformation of potential to kinetic energy). Cliff impact velocity
determines the amount of kinetic energy imparted to the water body at the time of
impact, which is important for wave generation. Finally, the number of fragments is
selected to demonstrate the effect of fragments of cliff upon impact with the water
surface on wave amplitude and runup height. Together, these parameters define the
energy budget from the state of rest to its release and then transfer to the water body to



181 its final stage as amplitude and runup. Since the wave velocity was not measured during

182 the experiments, a 2D numerical model was developed using Ansys-Fluent, and wave

183 velocity was measured; moreover, the results from the experiment were also cross-

184 validated.

185   Finally, based on experimental results, prediction modeling for the amplitude and

186 runup of water waves was carried out using multi-expression programming (MEP), and

187 a novel prediction model was developed for the water waves induced by rotational cliff

188 collapse, considering fragmentation of the cliff upon impact with the water surface.

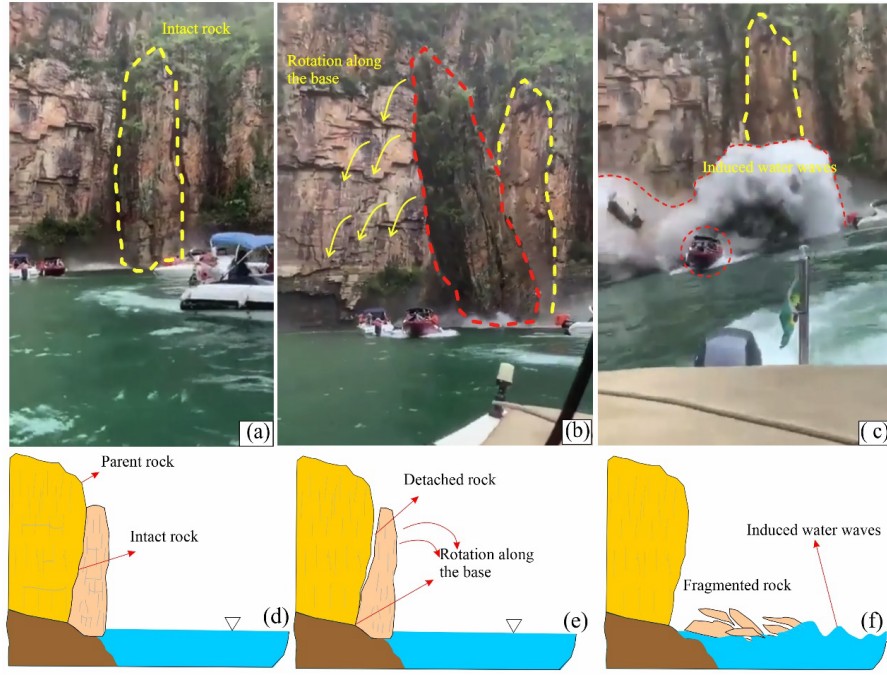

190 Fig. 1: (a, b, and c) waves induced by a cliff collapse in Lake Furnas, Brazil. (d, e, and
191    f) sketch diagram indicating the detachment and rotational fall process.

192 Table 1: Historical overview of the prediction models for wave amplitude

| Authors | Predictive model |
|---|---|
| Kamphuis and Bowering (1970) | $A_m = \left(\dfrac{v_s}{\sqrt{gh}}\right)^{0.7} \left(0.31 + 0.2\,log\left(\dfrac{l_s}{h^2}\right)\right) + 0.35e^{-0.08(x/h)}$ |
| Noda (1970) | $A_m = 1.32\dfrac{v_s}{\sqrt{gh}}$ |
| Huber and Hager (1997) | $\dfrac{H_m}{h} = 2 \times 0.88\,sin\,\theta\,cos^2\left(\dfrac{2\alpha}{3}\right)\left(\dfrac{\rho_s}{\rho_w}\right)^{0.25}\left(\dfrac{V}{wh^2}\right)^{0.5}\left(\dfrac{r}{h}\right)^{-\frac{2}{3}}$ |



| Fritz et al. (2004) | $A_m = 0.25 \left(\frac{v_s}{\sqrt{gh}}\right)^{1.4} \left(\frac{s}{h}\right)^{0.8}$ |
|---|---|
| Panizzo et al. (2005) | $\frac{Hm}{h} = 0.07 \left(\frac{T_s h^2}{ws}\right)^{-0.45} (\sin \alpha)^{-0.88} e^{0.6\cos\theta} \left(\frac{r}{n}\right)^{-0.44}$ |
| Heller (2007) | $A_m = \frac{4}{9}\left[F\left(\frac{s}{h}\right)^{1/2} \rho^{1/4} \left(\cos\frac{6\alpha}{7}\right)^2\right]^{4/5}$ |
| Mohammed and Fritz (2012) | $A_m = max\left(A_{C_1}, A_{C_2}\right)$ <br><br> $A_{c1} = 0.3F^{2.1}\left(\frac{s}{h}\right)^{0.6}\left(\frac{r}{h}\right)^{\left(-1.2F^{0.25}\left(\frac{s}{h}\right)^{-0.02}w-0.33/h\right)} \cos\alpha$ <br><br> $A_{c2} = 1.0FS^{0.8}\left(\frac{w}{h}\right)^{-0.4}\left(\frac{l}{h}\right)^{-0.5}\left(\frac{\gamma}{h}\right)^{-1.5F^{0.5}\left(\frac{w}{h}\right)^{-0.07}\left(\frac{w}{h}\right)^{-0.3}} \cos^2\alpha$ |
| Wang et al. (2016) | $A_m = 1.17F\left(\frac{sl}{bh}\right)^{0.25}\left(\frac{w}{b}\right)^{0.45}(\sin^2\alpha + 0.6\cos^2\alpha)$ |
| Li et al. (2023) | $A_m = 0.59\sqrt{\frac{2H(1-f\cot\alpha)}{h}}\left(\frac{swl}{h^3}\right)^{N-0.11}\left(\frac{x}{h}\right)^{-0.43}\cos^2\left(\frac{2}{3}\alpha\right)$ |

Note: *Note*: *l* is the landslide length; *s* is the landslide thickness; *w* is the landslide width; m is the landslide mass weight; *V* is the landslide volume; *H* is the landslide height; *b* is the river width; *h* is the still water depth; *x(r)* is the offshore distance from the bank slope; *α* is the slope angle; *θ* is the angular direction; *vs* is the impact velocity.

## 2. Research methodology

### 2.1 Model Preparation

The physical modeling for wave amplitude and runup induced by rotational cliff collapse was carried out in a three-dimensional water flume made up of plexiglass, as shown in Figs. 2 and 3. One end of the flume is vertical at 90°, whereas the other end is inclined and fixed at 30° (Fig. 3a and b). The flume is 0.55 m high, 0.5 m wide, and 1.4 m long along the base and 2.35 m long at the top. Furthermore, to measure the runup of induced water waves at various slope angles, two sliding rails were installed towards the inclined end at 45° and 60°. So, upon insertion of the gate at 45° and 60°, the top length of the flume was further reduced according to the Pythagoras theorem. To induce the rotational cliff collapse, a 0.55 m wide and 0.6 m high movable platform was prepared, which can move in the vertical direction and can also rotate about its axis. The flume was marked with a vertical scale to measure the water depth.





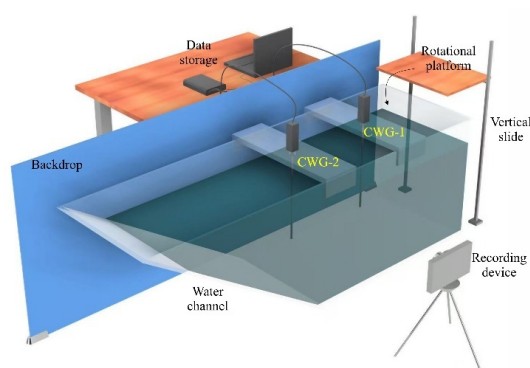


Fig. 2: Illustration of experimental setup including wave gauges, rotational platform,
recording, and data storage devices.

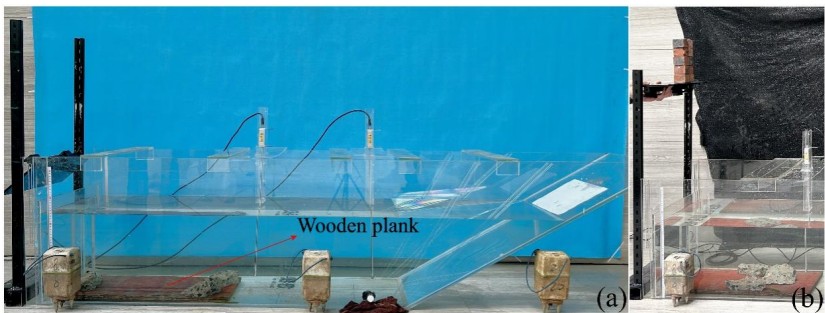


Fig. 3: Photographs of the setup, (a) Experimental flume, (b) platform for inducing
rotational cliff collapse.
**2.2 Test preparation and materials**
Physical experiments were carried out by varying the water depth, fall height,
number of fragments, bank slope angle, mass of falling rock, cliff height, and impact
velocity. The tests were carried out for three water depths, i.e., 0.34 m, 0.27 m, and 0.20
m, and three fall heights, i.e., 0.64 m, 0.44 m, and 0.245 m from the surface of the water
level. Furthermore, the number of blocks was also varied, i.e., 6, 10, and 12 blocks
having combined weights of 1.445 kg, 2.29 kg, and 2.82 kg, respectively. At the same
time, the impact velocity changed by changing the fall height. The wave runup was
measured by varying the bank slope angle, i.e., 30°, 45°, and 60°.
To replicate the field density of the rocks, red gutka bricks having a density of
around 2000 kg/m$^3$ were used. The single block had a dimension of 0.55 x 0.05 x 0.042



m. A combination of 6, 10, and 12 blocks of red gutka bricks were used to form a cliff
and measure the wave amplitude and runup of induced waves, the blocks were joined
together with the help of cement paste having water cement ratio W/C 0.8 and cured
for 2 hours Infront of an electric heater, such that the bond is weak enough that it
fragments at the joints upon impacting the water surface. The bonded blocks were
placed on the rotational platform at specific heights, i.e., 0.64 m, 0.44 m, and 0.245 m
from the water level, and were allowed to rotate under their own weight by pulling the
hinge, such that the placed block falls in the water following rotation motion along its
base Fig. 3 (b). To avoid the slippage of blocks and to replicate field conditions, fine-
grained bricks of the same material as the cliff were pasted on the rotational platform.
Furthermore, to reduce the impact of falling blocks on the base of the flume, a wooden
plank weighing 2.69 kg and dimensions 0.65m x 0.37m x 0.01 m was placed at the
point of impact inside the flume. Due to its large surface area and lighter density, it
tends to float in the flume, so two blocks of concrete weighing 3.58 kg were placed on
it, Fig. 3 (a).
The induced wave amplitude was measured by placing the wave measuring gauges
at a distance of 0.65 m and 0.135 m from the vertical face; the gauges were wired and
connected to the laptop. At the same time, the runup was measured manually with the
help of a scale by pasting a scaled paper on the slope. Furthermore, the experiments
were also recorded with the help of a high-resolution camera for verification purposes.
**2.3 Numerical Modeling**
Simulating multi-phase flows is challenging due to the constant deformation of the
liquid-gas interface. Various numerical methods have been developed to model these
flows, each offering unique advantages depending on the specific flow regime and
characteristics of interest. In this study, the Volume of Fluid (VOF) method is utilized
for its effectiveness in handling significant interface distortions and topological changes.
The VOF method offers superior mass conservation, which is critical in high velocity
impact conditions where liquid fragmentation and wave generation are significant
(Brackbill et al., 1992; Hirt and Nichols, 1981). Other approaches provide superior





accuracy in modeling interfaces and surface tension, but they struggle to manage
complex scenarios (Liu and Liu, 2010; Monaghan, 1994; Yang and Kong, 2018). Given
these trade-offs, the Volume of Fluid (VOF) method finds an optimal balance of
computational efficiency, interface tracking capability, and proven reliability for
modeling multiphase flow in the moderate-to-high velocity range relevant to this study.
Therefore, a two-dimensional numerical model of a cliff, having the same properties as
the experimental cliff mentioned previously, hitting the water surface is developed
using the VOF method to accurately capture the liquid-gas interface.
In this approach, a volume fraction ($\alpha$), ranging between 0 and 1, is applied across
the entire computational domain. A value of $\alpha = 1$ indicates a control volume filled with
liquid, while $\alpha = 0$ denotes a control volume filled with gas. The interface is represented
by values where $0 < \alpha < 1$. In the Volume of Fluid (VOF) method, the momentum
equation is solved across the entire computational domain, with the resulting velocity
field shared by all phases. To account for surface tension effects, a continuum surface
force (CSF) model is employed (Backbill et al., 1992). The normal vector $\boldsymbol{n}$ and
interface mean curvature $\boldsymbol{\kappa}$ are as follows, respectively:
$$n = \frac{\nabla \alpha}{|\nabla \alpha|} \tag{1}$$
and
$$\kappa = \nabla . \frac{\nabla \alpha}{|\nabla \alpha|} \tag{2}$$
The interface is maintained as sharp through the use of geometric reconstruction to
ensure its clarity. The volume fraction ($\alpha$) is discretized with the geo-reconstruct scheme,
while the convective terms in the momentum equation are handled using a second-order
upwind method. The PISO (Pressure-Implicit with Splitting of Operators) algorithm
was employed for pressure-velocity coupling, which is well-suited for transient flows.
Temporal discretization employs a second-order implicit scheme, and spatial gradients
are calculated using the Least Squares Cell-Based method.
To have an accurate simulation of the rotational motion of the cliff through the air-
water interface in a multi-phase flow environment, dynamic meshing was implemented
within the ANSYS Fluent framework. This approach facilitated the adaptation of the





computational mesh to accommodate the cliff's movement while maintaining the
integrity of the liquid-gas interface captured by the Volume of Fluid (VOF) method.
Dynamic meshing was critical for modeling the complex interactions between the
falling cliff and the surrounding air and water phases, allowing the mesh to deform and
adapt in response to the cliff's trajectory. In ANSYS Fluent, the dynamic meshing
strategy employed a combination of mesh deformation and local remeshing techniques
to handle the cliff's motion. Mesh deformation was applied to adjust the existing mesh
nodes smoothly as the cliff moved, preserving mesh quality in regions experiencing
moderate displacement. For areas near the cliff where significant deformation could
lead to poor mesh quality, local remeshing was utilized to regenerate mesh elements for
better numerical stability and accuracy. The smoothing and remeshing algorithms were
configured to maintain high mesh quality, with a skewness threshold set to prevent
excessive element distortion.
The rotational cliff collapse was simulated using an in-house user-defined function
(UDF). This UDF interfaced with ANSYS Fluent to dynamically update the rock's
position and velocity. To enhance computational efficiency, a dynamic mesh zone was
defined around the cliff, with a finer mesh resolution near its surface to capture the
sharp gradients in the flow field and interface dynamics. The mesh was gradually
coarsened away from the rock to reduce computational cost while maintaining
sufficient resolution in the far-field regions. The dynamic meshing process was
synchronized with the transient flow solver, using a time step size determined through
a time step independence study to balance accuracy and computational efficiency.
**2.4 Multi-expression programming**
The MEP model was developed for predicting wave amplitude and runup using
experimental data, as shown in Table 2. A dataset of 81 experiments was prepared by
alternately varying seven different parameters, and the results for wave amplitude and
runup were recorded. Furthermore, the data was divided into 70/30 ratios for training
and validation purposes before developing the model. The model starts working by
generating a random chromosome population, and it continues to generate the
chromosomes until a terminal condition is achieved, generating an optimal expression



from the data having input and output pairs over a certain number of generations, as
shown in Fig. 4.

Based on a binary tournament process, parents are selected and then undergo a

recombination process through a consistent crossover probability. This recombination
produces two more offspring. These offspring go through mutation, and if these
offspring perform better than the least fitting offspring in the current population, then
the better offspring replace them. The illustrations used by MEP are similar to the ones
used by C++ and Pascal compilers. The MEP chromosomes are comprised of numerous
genes combined using various mathematical operators such as addition (+), subtraction
(-), multiplication (x), and division (/), and these genes create expression trees (ETs)
(Cheng et al., 2020). Moreover, there are several parameters such as code length, sub-
population size and number, crossover probability, and other sets of various functions
involved in in generation of MEP code, and they also govern the overall performance
of the code. Among these parameters, the size of the population tells us about the
number of programs being generated, whereas an increase or decrease in subpopulation
size directly affects the complexity and computation time of the model. Moreover, the
length of the developed model is controlled by varying the code length parameter.
Table 2: Input parameters and corresponding wave amplitude and runup heights.

| S/No. | Water depth ($d$) | Drop height ($H$) | Fragments ($N_f$) | Angle ($\alpha$) | Cliff Mass($m$) | Cliff height($h$) | Velocity ($v$) | Amplitude ($A$) | Runup (m) |
|---|---|---|---|---|---|---|---|---|---|
| 1 | 0.34 | 0.245 | 6 | 30 | 1.445 | 0.12 | 2.19 | 0.0225 | 0.051 |
| 2 | 0.34 | 0.445 | 6 | 30 | 1.445 | 0.12 | 2.95 | 0.0230 | 0.056 |
| 3 | 0.34 | 0.645 | 6 | 30 | 1.445 | 0.12 | 3.56 | 0.0365 | 0.068 |
| 4 | 0.34 | 0.245 | 6 | 45 | 1.445 | 0.12 | 2.19 | 0.0370 | 0.045 |
| 5 | 0.34 | 0.445 | 6 | 45 | 1.445 | 0.12 | 2.95 | 0.0425 | 0.051 |
| . | . | . | . | . | . | . | . | . | . |
| . | . | . | . | . | . | . | . | . | . |
| . | . | . | . | . | . | . | . | . | . |
| 37 | 0.27 | 0.245 | 10 | 30 | 2.295 | 0.20 | 2.19 | 0.0431 | 0.116 |
| 38 | 0.27 | 0.445 | 10 | 30 | 2.295 | 0.20 | 2.95 | 0.0510 | 0.129 |
| 39 | 0.27 | 0.645 | 10 | 30 | 2.295 | 0.20 | 3.56 | 0.0685 | 0.141 |
| 40 | 0.27 | 0.245 | 10 | 45 | 2.295 | 0.20 | 2.19 | 0.0390 | 0.085 |
| 41 | 0.27 | 0.445 | 10 | 45 | 2.295 | 0.20 | 2.95 | 0.0523 | 0.102 |
| . | . | . | . | . | . | . | . | . | . |
| . | . | . | . | . | . | . | . | . | . |





| | . | . | . | . | . | . | . | . | . |
|---|---|---|---|---|---|---|---|---|---|
| 78 | 0.2 | 0.645 | 12 | 45 | 2.82 | 0.24 | 3.56 | 0.0733 | 0.146 |
| 79 | 0.2 | 0.245 | 12 | 60 | 2.82 | 0.24 | 2.19 | 0.0565 | 0.062 |
| 80 | 0.2 | 0.445 | 12 | 60 | 2.82 | 0.24 | 2.95 | 0.0636 | 0.083 |
| 81 | 0.2 | 0.645 | 12 | 60 | 2.82 | 0.24 | 3.56 | 0.0657 | 0.098 |

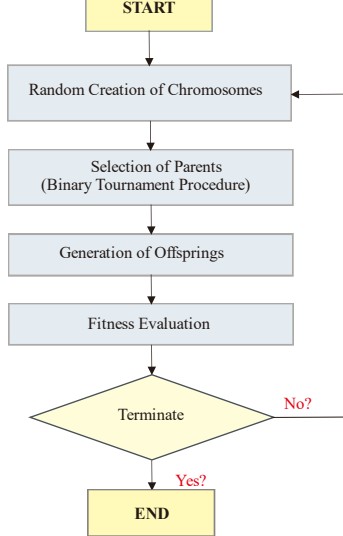


Fig. 4 MEP flowchart
**3. Results and discussions**
**3.1 Experimental results**
The experimental results of the wave amplitude and runup, induced by rotational
cliff collapse, reveal complex hydrodynamic processes. As shown in Fig. 5, the failure
is initiated by the rotational fall of the cliff, leading to a significant amount of impact
energy upon hitting the water surface. The impact induced a huge splash, which is
evident from Fig. 5 (b, e & h). It was observed that the shape of the splash also varies
with water depth for all the cases; higher water depths resulted in a mushroom-shaped
splash, i.e., broader on the top, as can be seen in Fig. 5(h). The observed phenomena
perfectly align with the basic concepts of fluid dynamics, which state that greater depths
absorb more impact energy compared to shallow waters. Shallow waters produced a
vertically elongated splash as can be seen in Fig. 5 (b & e). It can be observed that as
the depth decreases, the splash becomes more elongated, and this is due to the fact that
shallower depths intensify the upward momentum transfer, thus resulting in a more





elongated shape (Kubota and Mochizuki, 2009).

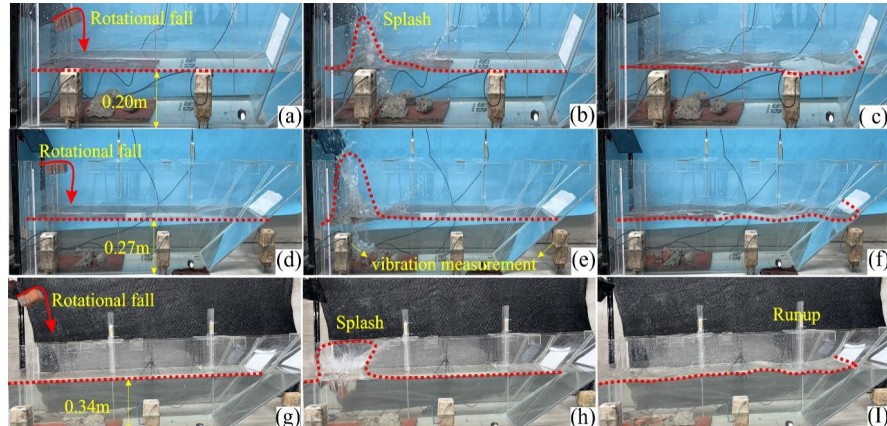


Fig. 5: A pictorial display of the experimental setup for various water depths, i.e., 0.20
m, 0.27m, and 0.34 m. (a, d & g) indicate rotational fall of the cliff, (b, e & h)
showing splash as a result of cliff impact, (c, f & I) formation and propagation of
induced wave and runup at various slope angles.

**3.1.1 Relation between energy and amplitude**

Further, the relationship between impact energy and wave response was also
investigated by establishing a dimensionless impact energy parameter (K.E/$\rho gh^3$).
Where *K.E* is the kinetic energy of the cliff, *ρ* is the density, and *h* is the water depth.
The negative quadratic coefficient in Fig. 6(a) indicates a nonlinear response, such that
at the start, the wave amplitude increases as the impact energy increases, but later it
decreases, due to reduced energy transfer at higher impact values. Moreover, the
coefficient of determination was found to be 77% indicating a good data fit.
Moreover, the results for the relative wave amplitude and wave energy were
analyzed for three water depths, i.e., 0.34 m, 0.27 m, and 0.20 m., as shown in Fig. 6(b).
The results indicate a strong correlation for all three cases, with coefficients of
determination around 0.96. The results indicate a direct relation between wave height
and energy, whereas the decreasing slope values with the increasing water depth
suggest that for deeper water the wave amplitude decreases at a slower rate with
increasing wave energy, thus highlighting the impact of water depth on the wave
dynamics, such that shallower water allows more amplification of waves for the same



energy level, and this is due to the increased non-linear interactions and enhanced
energy concentrations in shallower depths (Myrhaug and Lader, 2019).

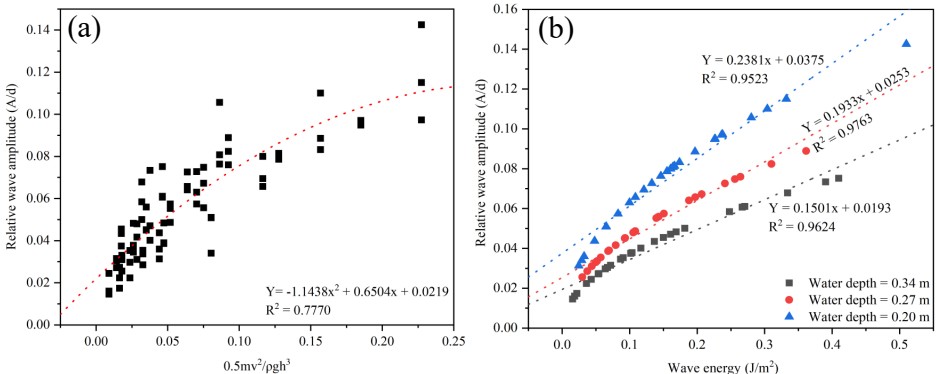


Fig. 6: (a) Impact energy vs relative wave amplitude, (b) Wave energy vs relative
wave amplitude

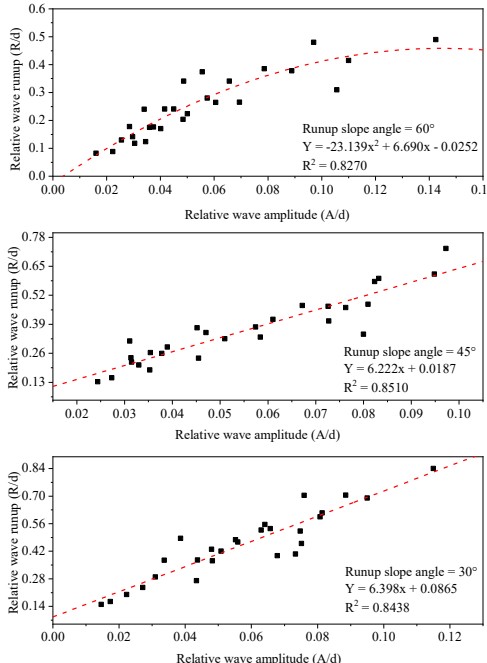


Fig. 7: Relative wave amplitude vs relative wave height.
The results for the relative wave height and runup for all three water depths and
three runup slope angles are shown in Fig. 7. The relationship indicates a strong
correlation between wave amplitude and runup for all three slope angles. The





decreasing line-slope values with increasing runup slope angle indicate that wave runup
increases at a slower rate for sharp slope angles compared to mild slopes. The trend
highlights the effect of slope angle on the runup. The result also indicates that the mild
slope angles help wave runup amplification, as they dissipate a very small amount of
energy, whereas steeper angles result in lower runup heights because of higher energy
losses (Wu et al., 2018).

**3.1.2 Impact Froude no vs Relative wave amplitude**

Fig. 8 indicates the relationship between the impact Froude number and relative
wave amplitude (A/d), under varying experimental conditions for the first gauge, i.e.,
near the impact zone. Since we are interested in the immediate response of the wave
influenced by the impact Froude number. The results indicate that as the water depth
decreases, the relative wave amplitude and impact Froude number increase, indicating
a reduction in the dissipation of impact energy, causing pronounced surface turbulence
and increased wave height. Additionally, the decreased water depth also increased the
value of the impact Froude number by reducing its characteristic velocity, resulting in
stronger wave generation upon impact. The calculations for Reynolds number for the
experiments resulted in very high values, thus indicating a strong turbulent flow, which
is also evident from Fig. 5, so viscous effects are very, very small and can be ignored,
thus indicating the Froude dynamics similarity. The experimental results indicate the
complex interaction between wave propagation, impact dynamics, and bathymetrical
effects in waves induced by rotational cliff collapse. Moreover, upon impact, the cliff
fragmentation distributes impact energy over a larger area of water, thus increasing
wave height by enhanced turbulence and water splashing effects.






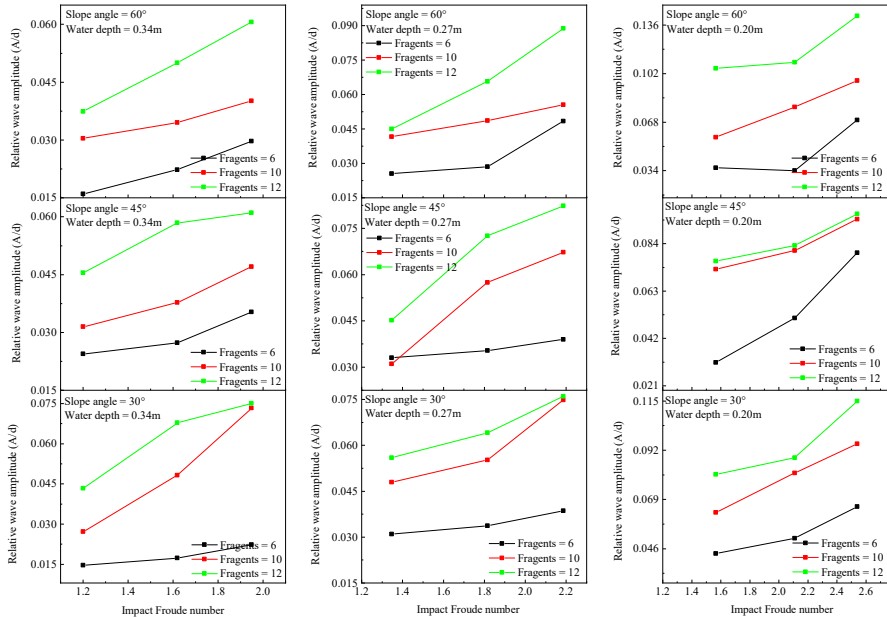

Fig. 8: Relationship between impact Froude number and relative wave height

### 3.1.3 Wave amplitude results

The results for the wave amplitude for various parameters are shown in Figs. 9, 10,
and 11. As mentioned earlier, two gauges were used to measure the induced wave
amplitude. Fig. 9 provides a detailed comparison of the wave amplitude recorded at
both gauges for 60°runup slope angle and a 0.445m fall height. It can be observed that
gauge-1, which is near to impact zone, has a higher relative amplitude compared to
gauge-2. Furthermore, the results for the relative wave amplitude against the
normalized time were also analyzed for all the water depths (0.20m, 0.27m, and 0.34m),
fall height (0.245m, 0.445m, and 0.645m), and cliff height (0.12m, 0.20m, and 0.24m).
The results indicate that the wave amplitude increases as the cliff height, impact
velocity, and number of fragments increase for all the water depths, as can be observed
in Fig. 10, thus demonstrating that the potential energy of the falling cliff plays a critical
role in the magnitude of the resulting wave.
Interestingly, comparing the wave amplitude induced by cliffs of various heights
falling from the same height revealed that the water depth and the wave have an inverse



relationship. As shown in Fig. 10 (a, b, and c), the average wave amplitude for various cliff heights and the same fall height of 0.245 m at 0.20m water depth is 26% more than the average wave amplitude induced by 0.27m water depth and 50% more than the 0.34m water depth wave amplitude. Similarly, Fig. 10 (d, e, and f) indicates that the average wave amplitude for 0.445m fall height at 0.20m water depth is 18% more than 0.027 m and 47% more than 0.34 m water depth, whereas, for 0.645m fall height wave amplitude induced by 0.20 m water depth is 25% more than 0.27m and 37% more than 0.34m water depth (Fig. 10 g, h & i), thus suggesting that the deeper water dissipates the impact energy more effectively, as the deep water have more mass available to absorb and redistribute the impact energy, compared to shallower water thus reducing the overall amplitude of the induced wave. Moreover, a similar trend was observed for the wave amplitude involving 45°and 60°runup slope angle.

Later on, we performed another experiment by using granular material of equivalent mass as of cliff and slid it on a 30° slope, for all the water depths, and amplitude of the induced wave was measured as shown in Fig. 11. Fig. 11(a) indicates that the wave amplitude for 0.20 m water depth and 1.445kg granular mass (equivalent to 0.12 m cliff height) was 15% more than 0.27m water depth and 65% more than wave amplitude induced by 0.34 m water depth. Whereas for 2.29kg and 2.82kg granular mass equivalent to 0.20 m and 0.24 m cliff height similar trend was observed as shown in Fig. 11 (b and c), thus indicating that as the water depth increases, the wave amplitude decreases for all the equivalent granular masses as happened in the case of cliff fall.

Furthermore, a comparison between the wave amplitude induced by a falling cliff and equivalent granular mass at various water depths indicates that the amplitude of the wave induced by an equivalent granular mass in 0.34m, 0.27m, and 0.20m water depth was on average 28%, 35% and 42% less than the wave amplitude induced falling cliff. The substantial difference in wave amplitude highlights the importance of energy transfer in wave formation. The falling cliff following a rotational motion imparts a more sudden and concentrated impact that allows an efficient energy transfer to water, leading to higher wave amplitudes. On the other hand, granular flows, being more deformable and flowing along a slope, result in gradual energy transfer over a wide area,



thus resulting in lower wave amplitudes. The results highlight that it's not only the total
impact energy that affects the behavior of the induced wave, but the mode of energy
transfer also plays a critical role (Mohammed and Fritz, 2012; Wunnemann and Weiss,
2015). Based on the experimental results for wave amplitude and runup induced by
rotational cliff collapse that fragments upon impact with the water surface, a novel
prediction model was prepared using multi-expression programming. The justifications
for the use of MEP have been well explained in the previous sections.

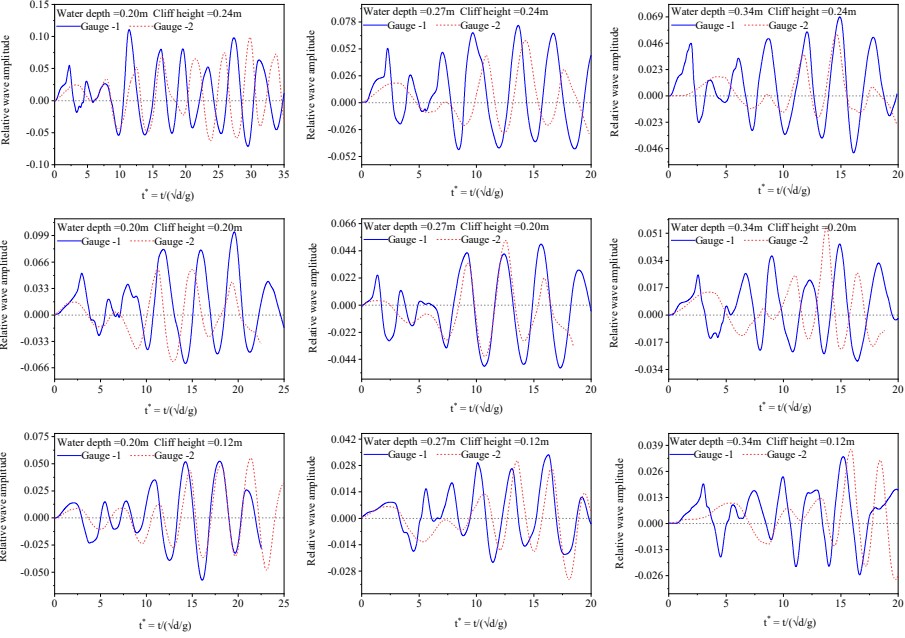


Fig. 9: A comparative display of the wave recorded at gauge 1&2 for a 60° slope
angle, and 0.445 m fall height.



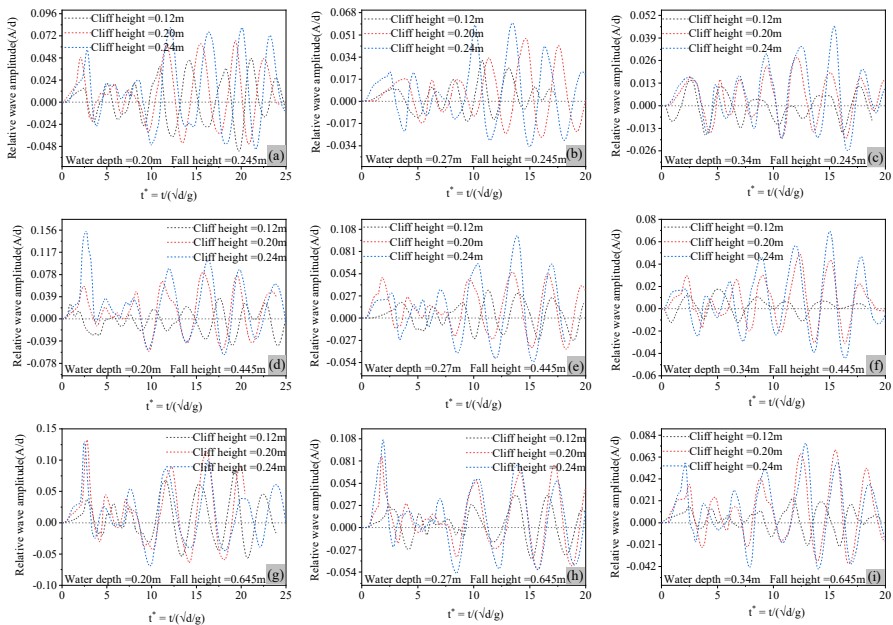


Fig. 10: Relative wave amplitude for various water depths, cliff height, and fall height
at 30°runup slope angle, (a, b&c) relative wave amplitude induced by 0.245m fall
height, (d, e&f) relative wave amplitude induced by 0.445m fall height, (g, h&i)
relative wave amplitude induced by 0.645m fall height.

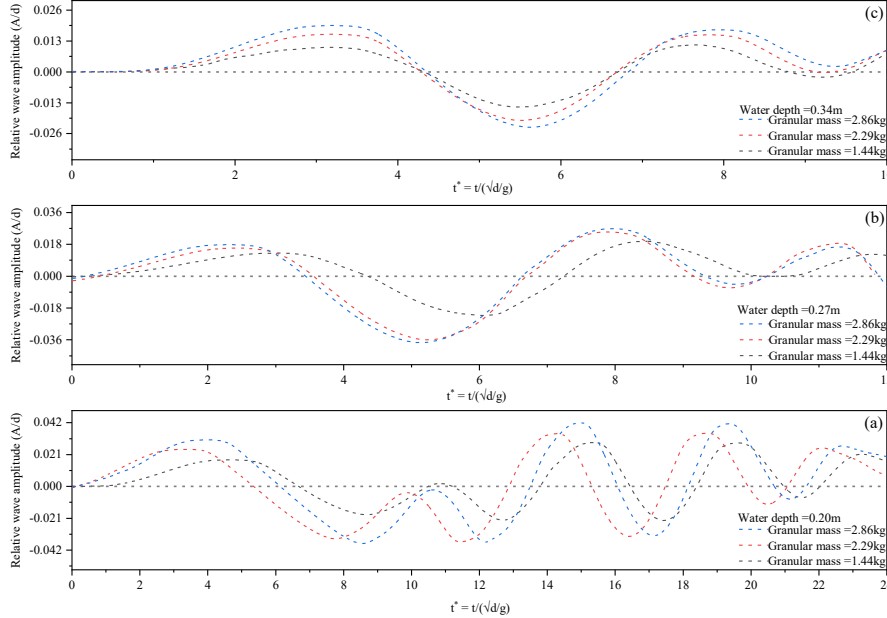


Fig. 11: Wave induced by equivalent granular mass



## 3.2 Numerical modeling results

The numerical simulations conducted in this study successfully captured key dynamic characteristics of the wave generated by the rotational cliff collapse, specifically the wave amplitude and wave runup, across a range of test cases. Moreover, the front velocity of the incident wave was also measured. The simulations were also focused on verifying the results obtained from the rotational cliff collapse in the experiments. To quantify the wave amplitude, runup, and velocity, a post-processing technique was employed. To establish the reliability of the wave front velocity measurements, the velocity was calculated at 5–7 distinct locations along the wave's propagation path and at multiple time steps during the simulation. This multi-point sampling approach minimized errors due to spatial and temporal variations. Fig. 12 shows a representative case of wave formation and propagation in a water tank at a depth of $d = 0.2$ m at various time frames.

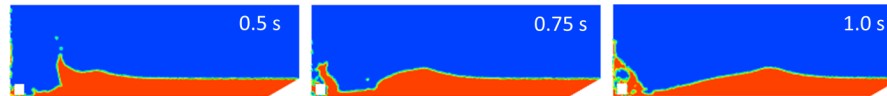

Fig. 12 Wave formation and propagation at water depth of $d = 0.2$ m at various time frames.

The wave amplitude was defined as the peak vertical displacement of the liquid surface relative to the undisturbed free surface level. Fig. 13 illustrates a representative case, depicting the wave front propagation.

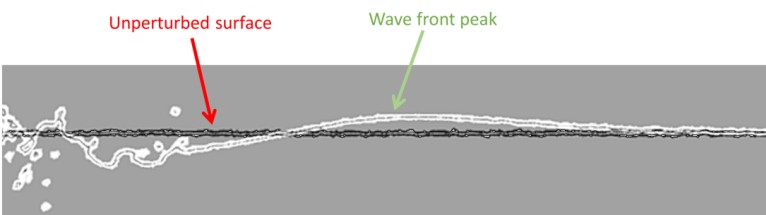

Fig. 13 Wave dynamics following a rotational cliff collapse in water depth $d = 0.34$ m. Stable liquid surface before impact (black line); wave front propagating to the right (white line).

To validate the results of simulations, we compared the results of the runup height with the experimental values. Table 3 presents the runup values for various





runup slope angles, i.e., 30°, 45°, and 60°, for a water depth of $d$=0.27m. The comparison of simulated values was performed at this depth, as it lies in the middle of the experimental test range of water depths. Numerical modeling results indicate that for a fixed water depth, the runup values consistently decrease as the runup slope angle increases from 30° to 60°. At a water depth of $d$=0.27 m, the runup decreases from 0.2 m at 30° to 0.17 m at 45°, and further to 0.11 m at 60°. This reduction is attributed to the changing momentum transfer dynamics with increasing slope angle. At less steep angles (closer to horizontal, e.g., 30°), the rock's momentum generates a stronger radial splash and greater upslope displacement of the liquid along the cliff. As the angle increases toward 60°, a larger component of the momentum is directed parallel to the cliff, reducing the vertical impulse. The experimental and numerical results agree well, and the difference lies within the acceptable range of 4-5%. The experimental results for the other two water depths also indicate similar behavior.

Table 3: Peak runup values along the various slope angles at a water depth of d=0.27m

| Depth $d$ (m) | Numerical-30º | Exp-30º | Numerical-45º | Exp-45º | Numerical-60º | Exp-60º |
|---|---|---|---|---|---|---|
| 0.27 | 0.20 | 0.19 | 0.17 | 0.16 | 0.11 | 0.102 |

Next, we measured the wave velocity through the numerical results, as it wasn't captured accurately through experimental images. Fig. 14 illustrates the simulated wave fronts at a time instant of $t$ =1 second following the impact of the solid rock on the liquid pool, for various water depths and a fixed slope angle of 30 degrees. These visualizations highlight the propagation of the waves from the impact zone. The slope angle was varied across simulations to assess its influence on wave characteristics. It was observed that changes in the slope angle induced only minor variations in both the wave front velocity and wave amplitude for a given pool depth. These perturbations were typically within 1–2% of the mean values. Consequently, to streamline the analysis and focus on dominant trends, the wave front velocity and height were averaged over the range of slope angles for each specific water depth.

However, variations in water depth exerted a pronounced effect on the wave




dynamics, leading to significant alterations in both the propagation velocity and amplitude of the generated waves. This depth-dependent behavior is quantified in Table 4, which presents the averaged results from the numerical simulations. For a shallow water depth of d=0.2 m, the average wave front velocity was computed as 1.48 m/s, with a corresponding average wave height of 0.11 m. As the pool depth increased to d=0.27 m, the velocity rose to 1.58 m/s, while the wave height decreased to 0.07 m. Further deepening to d=0.34 m yielded a velocity of 1.74 m/s and a reduced wave height of 0.06 m. These trends indicate an approximately linear increase in velocity with depth, accompanied by an inverse relationship for wave amplitude.

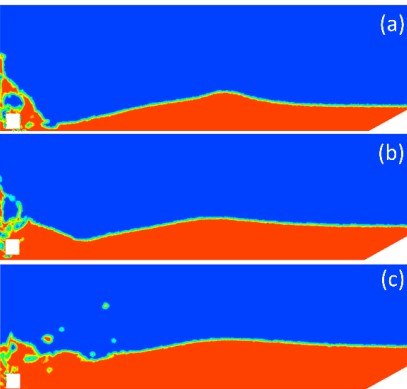

Fig. 14 Propagating wave fronts after the impact at time $t$ = 1 s for a slope angle of 30-degree. (a) d = 0.2 m, (b) d = 0.27 m, (c) d = 0.34 m.

481  The observed depth dependence can be rationalized through fundamental

482 principles of wave propagation in gravity-dominated, multi-phase flows. In the shallow

483 water regime, given that the pool depths (0.2–0.34 m) are comparable to or smaller than

484 the wavelengths of the generated waves, the phase velocity $c$ of long gravity waves

485 approximates $c \approx \sqrt{gh}$, where $g$ is the gravitational acceleration (9.81 m/s²) and $h$ is the

486 undisturbed water depth. This relation arises from the shallow water equations, where

487 hydrostatic pressure balance and negligible vertical acceleration dominate, leading to a

488 dispersionless incident wave speed that scales with the square root of depth.

489 Substituting the water depths yields theoretical velocities of approximately 1.40 m/s for

490 $d$=0.2 m, 1.63 m/s for $d$=0.27 m, and 1.83 m/s for $d$=0.34 m, which align closely with

491 the simulated values (discrepancies of 7–10% may stem from viscous dissipation, non-



hydrostatic effects near the impact zone, or spreading of the wave front). A comparative
analysis of the results is shown in Table 4.

Conversely, the decrease in wave amplitude with increasing water depth aligns with

energy conservation and volume displacement considerations in impact-generated
waves. The impact of rotational cliff collapse imparts a fixed kinetic energy and
displaces a finite volume of liquid, creating an initial cavity and subsequent outflow
that evolves into a propagating wave. In shallower pools, the displaced volume is
confined to a smaller cross-sectional area, resulting in greater vertical amplification to
accommodate the same mass redistribution. For deeper water depths, the energy is
distributed over a larger water column, diluting the surface perturbation and yielding
lower amplitudes. The trends observed in the numerical simulations for water waves
induced by rotational cliff collapse are in good agreement with theoretical and
experimental results, indicating that water depth has a direct effect on the wave velocity
and an inverse effect on the wave amplitude and runup.
Table 4**:** The average wave propagation velocity and amplitude for various water depths.

| Water depth $d$ (m) | Avg. wave velocity $v$ (m/s) | Theoretical wave velocity $c$ (m/s) | Wave amplitude (m) |
|---|---|---|---|
| 0.2 | 1.48 | 1.40 | 0.11 |
| 0.27 | 1.58 | 1.63 | 0.07 |
| 0.34 | 1.74 | 1.83 | 0.06 |


**3.3 MEP model results**

The purpose was to develop a precise model for wave amplitude and runup induced

by rotational cliff collapse. The predicted model is a function of seven variables, i.e.,
water depth, fall height, cliff mass, impact velocity, cliff height, runup slope angle, and
number of fragments, and can be described as follows,
$Wave\ amplitude\ and\ runup\ = f\left(d, H, m, v, h, \alpha, N_f\right)$ (3)

The relation among the parameters was evaluated using Pearson's correlation to

analyze the multicollinearity and interdependency between the parameters, as they can



obscure the interpretation of the developed model. The model was developed by
splitting the data into two subsets, i.e., training (70%) and testing (30%). The
randomization was done by MEP itself. Following the criteria, 70% of the data, i.e., 57
data points, were taken as training data, whereas 30% of the data, i.e., 24 data points,
were considered for validation of the model. The mathematical expression for MEP is
obtained by solving the C++ code and representing it as per optimized hyperparameter
settings, as shown in Table 5. The prediction model for wave amplitude and runup was
developed by analyzing the MEP code in MATLAB, as shown in Equations 4 and 5.
Table 5: Parametric settings of the MEP algorithm for wave amplitude and runup

| Sr.No. | Parameters | Wave amplitude | Wave runup |
|---|---|---|---|
| 1 | Number of sub-populations | 125 | 85 |
| 2 | Sub-population size | 115 | 75 |
| 3 | Crossover probability | 0.85 | 0.60 |
| 4 | Code length | 35 | 25 |
| 5 | Tournament size | 30 | 10 |
| 6 | Mutation probability | 0.085 | 0.06 |
| 7 | Number of generations | 250 | 120 |
| 8 | Crossover type | Uniform | Uniform |
| 9 | Error measure | Mean absolute error | Mean absolute error |
| 10 | Problem type | Regression | Regression |
| 11 | Function set | +, -, x, /, ^ | +, -, x, /, ^ |
| 12 | Terminal set | Problem Input | Problem Input |
| 13 | Operators | 0.5 | 0.5 |
| 14 | Simplified | Yes | Yes |
| 15 | Variables | 0.5 | 0.5 |
| 16 | Random seed | 0 | 0 |
| 17 | Constants | 0 | 0 |

$$Wave\ amplitude\ A = d^{\left(\frac{\alpha}{d\left(d+N_f+m\right)}\right)} + \frac{2vh^2}{m+N_f+d\left(d+N_f+m\right)} + 2vhd^{\left(\frac{\alpha}{d\left(d+N_f+m\right)}\right)} \qquad (4)$$
$$Wave\ runup\ R = \frac{A\left(h+\left(A\cdot\left(d-\frac{B}{\alpha}\right)\right)^{B/\alpha}\right)^{A}\cdot B}{\alpha} \qquad (5)$$
$A = v + h^d$
$B = v + m + h^d$
Whereas $d$ is the water depth (m), $m$ is the mass of the cliff (kg), $v$ is the
impact velocity (m/s), $h$ is the cliff height (m), $\alpha$ is the runup slope angle, and $N_f$ is





the number of fragments.

**3.3.1 Prediction performance of the developed model**

The robustness of the proposed model was evaluated by comparing it with well-
established statistical indices, i.e., mean absolute error (MAE), root mean square error
(RMSE), correlation coefficient (Cr), Nash–Sutcliffe efficiency (NSE), and
performance index (PI). The indices can be represented by equation (6-10) (Alavi et al.,
2010; Khan et al., 2022).

$$MAE = \frac{\sum_{i=1}^{n}|e_i - p_i|}{n} \tag{6}$$

$$RMSE = \frac{\sum_{i=1}^{n}(e_i - p_i)^2}{n} \tag{7}$$

$$NSE = 1 - \frac{\sum_{i=1}^{n}(e_i - p_i)^2}{\sum_{i=1}^{n}(e_i - \bar{e}_i)^2} \tag{8}$$

$$PI = \frac{RRMSE}{1+R} \tag{9}$$

$$R^2 = \left(\frac{\sum_{i=1}^{n}(e_i - \bar{e}_i)(p_i - \bar{p}_i)}{\sum_{i=1}^{n}(e_i - e_i)^2 \sum_{i=1}^{n}(P_i - \bar{P}_i)^2}\right)^2 \tag{10}$$

Whereas, $\bar{e}_i$ and $\bar{p}_i$ are the average values of the experimental and predicted
results, and $e_i$ and $p_i$ are $i_{th}$ values of the modeled and predicted results, for *n* total
samples. It is good to consider the error indices while analyzing the predictive
capability of complex models. The wave runup model demonstrated a robust
performance for both training and testing datasets. The lower values of RMSE and
MAE indicate little deviation from experimental values, while RSE and RMSE values
confirm lower normalized error, as shown in Table 6. The higher values of NSE and Cr
further validated the model reliability for the training phase. Whereas for the validation
dataset, i.e., the unseen data model displays even stronger performance with lower
RMSE and MAE values compared to the training dataset. Moreover, higher Cr and
lower performance index values highlight enhanced model efficiency. This suggests
that the model works well for unseen data, making it suitable for predicting the wave
runup induced by rotational cliff collapse (Gardezi et al., 2024).
The predictive performance of the wave amplitude model in the case of training
data demonstrated a strong correlation with high R$^2$ values and low RMSE and MAE





values corresponding to 13.14% relative error, thus suggesting a good agreement
between experimental and predicted values, as shown in Table 6. The higher NSE and
$C_r$ values further confirmed the model's reliability for the training dataset with minimal
systematic bias. When the model was exposed to unseen data, it still maintained
reasonable accuracy with an $R^2$ value of 0.78. Though the values of error matrices, i.e.,
RMSE, MAE, and RRMSE, are a bit higher than the training data set, this is expected
due to inherent generalization challenges. Similarly, the higher NSE and $C_r$ values,
though lower than the training dataset, indicate consistent predictive performance of
the wave amplitude model with little increase in bias. Overall model exhibited strong
predictive performance in the training and testing phase, with a little expected decline
in the validation phase.
Table 6: Performance index values for MEP-based velocity prediction model.

| Performance parameters | Wave Amplitude | | Wave Runup | |
|---|---|---|---|---|
| | Training data | Validation data | Training data | Validation data |
| RSQ | 0.8823 | 0.7811 | 0.8748 | 0.9691 |
| RMSE | 0.00178 | 0.0025 | 0.01327 | 0.00617 |
| MAE | 0.00135 | 0.00176 | 0.0108 | 0.00504 |
| RSE | 0.1180 | 0.2439 | 0.1306 | 0.0312 |
| RRMSE | 0.1314 | 0.1594 | 0.1472 | 0.0660 |
| P. index | 0.0698 | 0.0908 | 0.076 | 0.0333 |
| NSE | 0.8819 | 0.7560 | 0.8693 | 0.9687 |
| $C_r$ | 0.9393 | 0.8829 | 0.9353 | 0.9844 |

Previously, scientists have also used the slope of the regression line as a
performance indicator for AI models, thus representing a correlation between
experimental and predicted results. Fig. 15 (a & b) shows the regression line for our
wave amplitude and runup model. For wave amplitude, the slope value for the training
data set is 0.88, which is adequate, and 0.78 in validation, which is still greater than the
minimum value of 0.7; it can happen as the model involving numerous parameters and
complex phenomena usually performs slower for the unseen data(Yarkoni and Westfall,
2019). Whereas, for wave runup, the model performed very well for both training and
validation data sets with an $R^2$ value of 0.87 and 0.96, respectively.





The accuracy of the proposed model can also be checked using residual error plots,
which are obtained by subtracting experimental and predicted values (Alavi et al., 2013).
The results indicate that the amplitude model has minimum and maximum values of -
0.004 m and 0.0065 m, as shown in Fig. 16 (a), whereas for wave runup the minimum
and maximum values are -0.01875 and 0.024 (Fig. 16b). Moreover, it can also be
observed that error values are populated along the x-axis, therefore, showing low error
frequency, and accuracy of both the models.

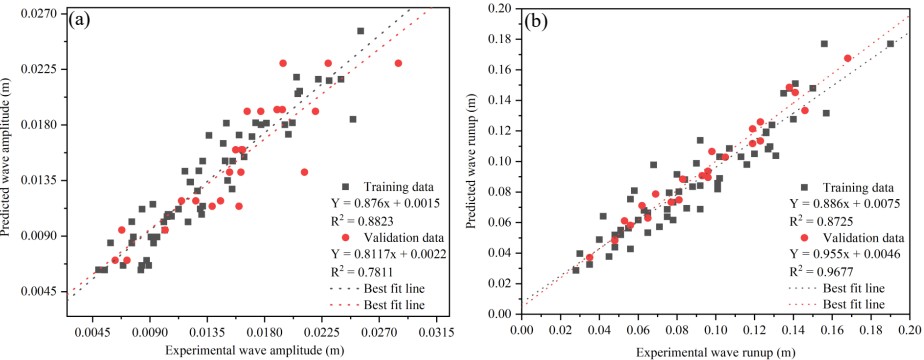

Fig. 15 Tracing the experimental results by predicted values, (a) wave amplitude and (b) wave runup

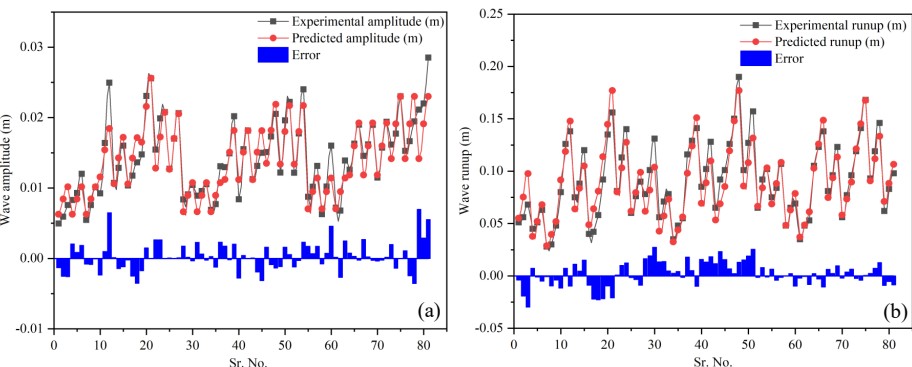

Fig. 16 Indicating error values between experimental and predicted model (a) wave amplitude, and (b) Wave runup

**3.2 Validation of the developed model**

The validation of the proposed model is an important feature in predictive modeling.
It has been observed that sometimes the model performs very well for training data sets,



but fails to perform during the validation stage for unseen data. So, the developed
prediction model was further validated by conducting the sensitivity and parametric
analysis for both the wave amplitude and runup.
**3.2.1 Sensitivity analysis**
Sensitivity and parametric analysis play a vital role in determining the robustness
of the proposed model. The sensitivity analysis (SA) of the developed prediction model
for the entire dataset tells us how sensitive the model is to any changes in input
parameters. So for an independent parameter $Y_i$ the SA can be calculated using
equations 11 and 12, which indicates that for any parameter, the values were varied
between two extremes and others were constant to their average, and the outcome was
found in the form of $Y_i$, and then the same process was repeated for all the remaining
parameters.
$R_k = f_{max}(Y_k) - f_{min}(Y_k)$ $\hspace{5cm}$ (11)
$Relative\ Importance\ SA\ (\%) = \dfrac{R_k}{\sum_n^{j=1} R_j} \times 10$ $\hspace{3cm}$ (12)
Whereas, $f_{max}(Y_k)$ and $f_{min}(Y_k)$ represent the minimum and maximum values
of the model-based results grounded on the *kth* domain of the input parameters in the
above equation. Fig. 17 (a & b) shows the results of the sensitivity analysis of the
developed prediction model for the wave amplitude and runup. Figure 17 (a) indicates
that the wave amplitude is greatly influenced by the height of the cliff (*h*) and has an
effect of almost 51%. The water depth (*d*) contributes 4.36% to wave amplitude, cliff
mass (*m*) contributes 4.69%, and impact velocity (*v*) and number of fragments ($N_f$)
contributes 18% and 22% to the induced wave amplitude. Whereas the fall height (*H*)
and runup slope angle (*α*) do not affect the wave amplitude. Since the impact velocity
parameters have already catered for the fall height that's why it is not visible in the
proposed model. The model tells us that impact velocity, cliff height, and number of
fragments contribute approximately 90% to the wave amplitude induced by the
rotational fall of the cliff. It can be concluded that the effect of $h > N_f > v > m >$
$d$ on the induced wave amplitude.
Similarly, the sensitivity analysis of wave runup (Fig. 17b) indicates that runup is





greatly influenced by bank slope angle (α) and has an effect of 34%. Impact velocity
(*v*) contributes 25.3%, cliff mass (*m*) 20.3%, cliff height (*h*) 13.3%, and water depth
(*d*) contributes around 7% to wave runup. Whereas, the number of fragments and fall
height that have already been catered in impact velocity don't contribute to wave runup.
This suggests that wave runup is primarily governed by coastal geometry, i.e., bank
slope angle and cliff height, and hydrodynamic forces, i.e., impact velocity, whereas
water depth contributes a little to wave runup. It can also be concluded as the effect of
$\alpha > v > m > h > d$ on the induced wave amplitude.

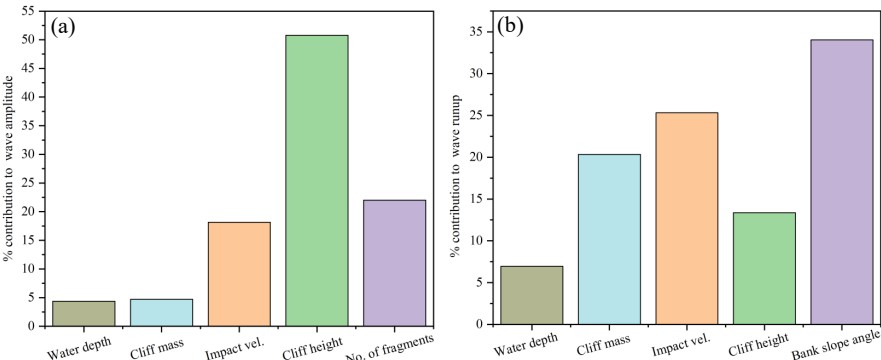


Fig. 17 Sensitivity analysis of the MEP-based wave amplitude and runup model
**3.2.2 Parametric Analysis**
Parametric analysis results for the input parameters for the wave amplitude used in
this study are displayed in Fig. 18. The parametric analysis indicates that wave
amplitude decreases as the water depth, number of fragments, and cliff mass increase,
whereas it increases with the increase in cliff height and impact velocity. These trends
are in line with the fundamental physics principles (Bougouin et al., 2020; Lipiejko et
al., 2023)– deep waters dissipate more energy, and greater impact velocities and larger
cliff heights impart more kinetic and potential energies to water bodies for wave
generation. Whereas, the inverse relation between the number of fragments and wave
amplitude proposes a potential threshold effect in which initial fragmentation
contributes to wave formation, whereas excessive fragments contribute to energy
dissipation owing to increased turbulence. The sensitivity analysis further quantified
the effect of these parameters, classifying cliff height as a major contributing factor in



wave amplitude variations, followed by impact velocity, number of fragments, water
depth, and mass of cliff. The strong influence of cliff height indicates its direct effect in
determining the potential energy for wave generation. Moreover, the larger sensitivity
value of fragments regardless of their inverse parametric relation shows a complex
relation, where fragment count plays a considerable but context-dependent role in wave
generation and propagation. The dominance of cliff height, impact velocity, and
fragment count suggests that these parameters should be prioritized in future prediction
models. These findings are important for developing predictive models for wave
generations due to rotational cliff collapse.
The developed model for wave amplitude provides valuable insights into
fundamental physics governing wave formation and propagation induced because of
rotational cliff collapse. The strong height dependence of the model confirms the
classical principle of conservation of potential energy, whereas the fragment count
dependence reveals energy partitioning mechanisms. The results of performance
indices and sensitivity, and parametric analysis increase our understanding of how
geometric and dynamic characteristics govern the wave characteristics, with relevance
to hazard assessment and disaster mitigation in coastal regions prone to cliff collapse
following rotational motion.
The results of the parametric analysis for wave runup are presented in Fig. 19. It
can be observed from Fig. 19 (a & e) that as the water depth and bank slope angle
increase, the wave runup decreases, due to energy dissipation and different wave
breaking dynamics. Conversely, as the cliff mass, cliff height, and impact velocity
increase, the wave runup increases, as greater kinetic energy and inertia impart greater
uprush. Notably, all the parameters present a strong correlation with the runup (more
than 97%), highlighting their statistical significance. The results agree with the general
physics laws, where mild slopes and larger impact forces result in higher runups,
whereas deep waters attenuate wave energy.
An important observation from parametric analysis of wave amplitude and runup,
as shown in Fig. 18b, and 19c, indicates that cliff mass represents a nonlinear relation
with wave amplitude and a linear relation with runup. This is due to the fact that the



variations in wave amplitude are governed by a nonlinear energy dissipation, where
hydrodynamic forces follow a quadratic dependence on the velocity. In the case of light
cliff collapses, the dynamic responses result in complex absorption and distribution,
whereas heavier cliff collapses promote wave reflection along with nonlinear effects of
wave breaking and splash-induced turbulence, as can be observed in Fig. 5 (b, e&h).
Conversely, the wave runup exhibits a linear relation with cliff mass, and this is due to
the law of conservation of momentum, such that the resisting inertial force is directly
proportional to cliff mass. The greater resistance to motion of heavier cliffs allows more
energy to be conserved and utilized for higher wave runups before dissipation. The
main difference between the two trends is that the wave amplitude is controlled by
localized energy losses, whereas runup is governed by bulk momentum transfer rather
than localized losses.

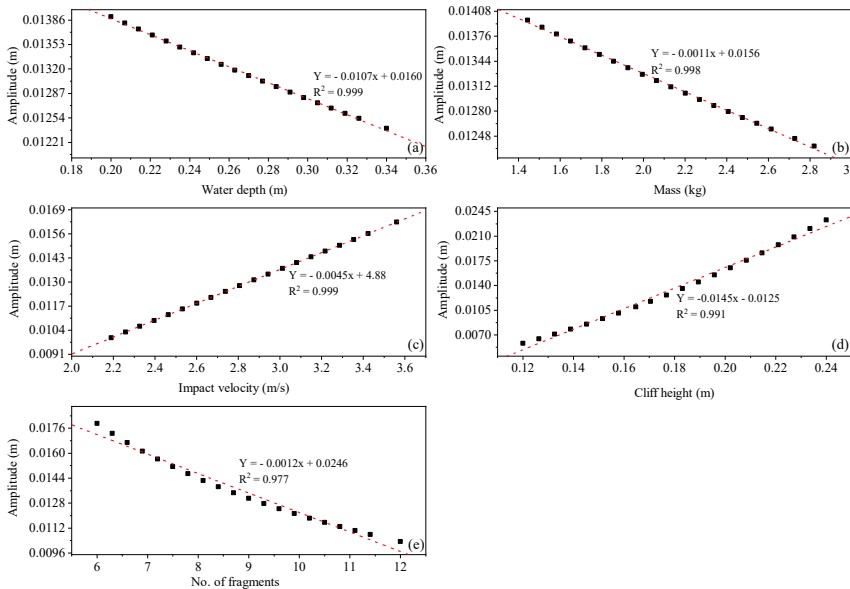


Fig. 18 Parametric analysis for wave amplitude (a) water depth, (b) cliff mass, (c)

impact velocity, (d) cliff height, (e) number of fragments.



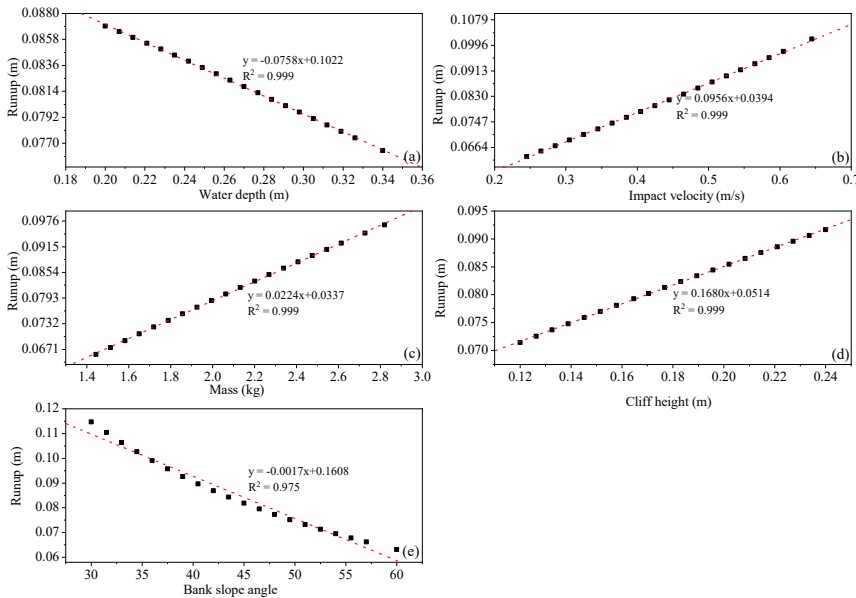

Fig. 19 Parametric analysis for wave runup (a) water depth, (b) impact velocity, (c) mass of the cliff, (d) cliff height, (e) bank slope angle.

## 4. Conclusions

While designing wave protection structures along the banks of reservoirs, it is common to use the empirical relations developed for granular flows, i.e., landslides and avalanches, to predict the amplitude and runup of the waves. However, the waves induced by various types of slides behave differently and should be treated accordingly. The dynamics of the waves induced by falling cliffs are entirely different from the waves induced by continuous granular flows. Similarly, the dynamics of the waves induced by falling cliffs following different types of motion (translational, rotational) are also different. This study aimed to develop a novel wave amplitude and runup prediction model for waves induced by rotational fall of the cliff using a combination of seven governing parameters, and then compare it with the dynamics of the wave induced by continuous granular flows. Based on the results and discussions, the study concludes as follows,

1. It was concluded that the shape of the induced splash depends on water depth; increased depth forms a mushroom-shaped splash, whereas shallow water forms a



vertically elongated splash. Moreover, shallow water allows more amplification of
waves for the same energy level compared to deep water.
2.  The effect of viscous forces is very, very small and can be ignored, since the
Reynolds number for all the experiments is very high, thus leaving the Froude
number as the best possible dynamic scaling factor. It is concluded that the Froude
number also increases as the water depth decreases.
3.  The study concludes that wave amplitude is greatly influenced by cliff height,
impact velocity, and the number of fragments. For all the cases, the deep water
dissipated more energy compared to shallow waters, thus resulting in lower
amplitudes.
4.  The amplitude of the wave induced by equivalent granular mass is lower than the
waves induced by rotational cliff collapse, thus concluding that the mode of energy
transfer to the water body plays a critical role in wave dynamics.
5.  A second-level validation of the developed model was performed by conducting
sensitivity and parametric analysis. It is concluded that the amplitude is highly
sensitive to any change in cliff height, impact velocity, and number of fragments.
In contrast, runup greatly depends on runup slope angle, impact velocity, and mass
of the cliff.
**Funding**
This work was supported by the National Key R&D Program of China (Grant No.
2023YFC3007001), the China Postdoctoral Science Foundation (Grant no.
2024M762420), the National Natural Science Foundation of China (Grant no.
42202312), and the Fundamental Research Funds for the Central Universities.
**Declarations**
Competing interests: The authors declare no competing interests.

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
