# Peer review of "Predicting the amplitude and runup of the water waves induced by rotational cliff collapse, considering fragmentation"

_EGUsphere, 2025_

## Author Comment (AC4)

**Response to the comments**

We are thankful to Reviewer (RC2) for their time and valuable feedback on our manuscript. We have carefully addressed and justified all the observations made in the review. Text in red color highlights the changes made in the revised manuscript.

1. Please ensure the **abstract** is short but reflects the approach, results, and conclusions correctly and concisely. Please check the keywords and highlights to ensure they are appropriate and complete. Highlights should be very brief and to the point and attractive to the readers of this journal. Kindly rearrange the keywords according to alphabetical order.

**Response:** Following the suggestion, the abstract has been modified, and keywords have been arranged according to alphabetical order. "Cliff collapses in small lakes and reservoirs induce powerful waves, threatening the offshore infrastructure. Unlike previous studies on waves induced by granular slide, this study experimentally and numerically investigates the waves induced by rotational cliff collapse, whereby the cliff fragments upon impact with the water surface, and determines the wave amplitude, runup, and energy transfer mechanics. Results indicate that as the water depth decreased, the impact Froude number and relative wave amplitude increased, wave velocity decreased, and splash showed greater elongation. The numerical modelling results also confirmed the experimental trends. Moreover, compared to an equivalent amount of granular mass sliding down a 30° slope, rotational cliff collapse produced 28-42% higher wave amplitudes due to the acute impact that transfers energy more efficiently. Machine learning based prediction models were subsequently developed to predict the wave amplitude and runup. The prediction models performed well both in the training and testing stages, with high R2 values, and were validated via established statistical indices, sensitivity, and parametric analysis. The prediction models highlighted a cumulative 90% contribution of impact velocity, cliff height, and the number of fragments on the wave amplitude. In comparison, runup is greatly influenced by bank slope angle, impact velocity, cliff mass, and height. The experimental results and developed prediction models can provide the basis for understanding the rotational cliff collapse-induced waves and can help with disaster mitigation and risk assessment by effectively predicting the wave amplitude and runup.

Keywords: Cliff fragmentation; landslide tsunami; prediction models; rotational cliff collapse;

**wave amplitude, and runup."**

2. The paper lacks a comprehensive background and literature review section in a **tabulated form**. To enhance the significance of the study, the authors should include an in-depth review of related literature. To enhance the clarity and comprehensiveness of the introduction section, the authors are kindly requested recommended to include a pertinent table.

**Response:** We thank the reviewer for the suggestion. A comprehensive literature summary on the existing prediction models for the water wave amplitude and runup is already included in the original manuscript as Table 1. This table does not duplicate the literature that has been discussed in the text; rather, it provides a detailed summary of the key studies related to prediction modelling of water waves, thus highlighting the most relevant and fundamental work in the field of wave mechanics.

| Authors                       | Predictive model                                                                                                                                                                                                                                                                                                                                                                                                           |
|-------------------------------|----------------------------------------------------------------------------------------------------------------------------------------------------------------------------------------------------------------------------------------------------------------------------------------------------------------------------------------------------------------------------------------------------------------------------|
| (Kamphuis and Bowering, 1970) | $A_m = \left(\frac{v_s}{\sqrt{gh}}\right)^{0.7} \left(0.31 + 0.2 \log\left(\frac{l_s}{h^2}\right)\right) + 0.35e^{-0.08(x/h)}$                                                                                                                                                                                                                                                                                             |
| (Noda, 1970)                  | $A_m = 1.32 \frac{v_s}{\sqrt{gh}}$                                                                                                                                                                                                                                                                                                                                                                                         |
| (Huber and Hager, 1997)       | $\frac{H_m}{h} = 2 \times 0.88 \sin \theta \cos^2 \left(\frac{2\alpha}{3}\right) \left(\frac{\rho_5}{\rho_w}\right)^{0.25} \left(\frac{V}{wh^2}\right)^{0.5} \left(\frac{r}{h}\right)^{-\frac{2}{3}}$                                                                                                                                                                                                                      |
| (Fritz et al., 2004)          | $A_m = 0.25 \left(\frac{v_s}{\sqrt{g_h}}\right)^{1.4} \left(\frac{s}{h}\right)^{0.8}$                                                                                                                                                                                                                                                                                                                                      |
| (Panizzo et al., 2005)        | $\frac{H_m}{h} = 0.07 \left(\frac{T_s h^2}{ws}\right)^{-0.45} (\sin \alpha)^{-0.88} e^{0.6\cos^{\theta}} \left(\frac{r}{n}\right)^{-0.44}$                                                                                                                                                                                                                                                                                 |
| (Heller, 2007)                | $A_{m} = \frac{4}{9} \left[ F\left(\frac{s}{h}\right)^{1/2} \rho^{1/4} \left(\cos\frac{6\alpha}{7}\right)^{2} \right]^{4/5}$                                                                                                                                                                                                                                                                                               |
| (Mohammed and Fritz, 2012b)   | $A_{m} = \max(A_{c_{1}}, A_{c_{2}})$ $A_{c_{1}} = 0.3F^{2.1} \left(\frac{s}{h}\right)^{0.6} \left(\frac{r}{h}\right)^{\left(-1.2F^{0.25}\left(\frac{s}{h}\right)^{-0.02}w - 0.33/h\right)} \cos\alpha$ $A_{c_{2}} = 1.0FS^{0.8} \left(\frac{w}{h}\right)^{-0.4} \left(\frac{l}{h}\right)^{-0.5} \left(\frac{\gamma}{h}\right)^{-1.5F^{0.5}\left(\frac{w}{h}\right)^{-0.07}\left(\frac{w}{h}\right)^{-0.3}} \cos^{2}\alpha$ |
| (Wang et al., 2016b)          | $A_m = 1.17F\left(\frac{sl}{bh}\right)^{0.25} \left(\frac{w}{b}\right)^{0.45} (\sin^2\alpha + 0.6\cos^2\alpha)$                                                                                                                                                                                                                                                                                                            |
| (Li et al., 2023b)            | $A_m = 0.59 \sqrt{\frac{2H(1 - f \cot \alpha)}{n}} \left(\frac{swl}{n}\right)^{N-0.11} \left(\frac{x}{n}\right)^{-0.43} \cos^2\left(\frac{2}{3}\alpha\right)$                                                                                                                                                                                                                                                              |

Note: l is the landslide length; s is the landslide thickness; w is the landslide width; m is the landslide mass weight; V is the landslide volume; H is the landslide height;  $T_s$  time for motion of slide, b is the river width; h is the still water depth; x(r) is the offshore distance from the bank slope; a is the slope angle;  $\theta$  is the angular direction;  $v_s$  is the impact velocity.

3. I am curious about the 'weak bond' as it is a critical parameter. It is recommended to provide a quantitative measure of bond strength. For example, what strength did you achieve by using 0.8 w/c and curing it for two hours? And how do you compare it with the inertial stresses upon impact with the water surface? This would allow others to reproduce the phenomenon and get results."

Response: We acknowledge the reviewer's observation, we have provided a detail on the bond strength in the revised manuscript. "To ensure the weak bond strength, several trials for bond strength were carried out after a curing period of 2 hours, and it was found to be in the range of 0.42-0.5 MPa. In contrast, the inertial stresses at the time of impact were several times higher, such that they caused the fragmentation of the cliff. This condition was purposely designed to imitate naturally fractured cliff materials, confirming that the structure fragmented primarily along the joints upon impact with the water surface, consistent with field observations of rotational cliff collapses".

4. "The negative quadratic coefficient in Fig. 6(a) indicates a nonlinear response, such that at the start the wave amplitude increases as the impact energy increases, but after a certain value it decreases, due to reduced energy transfer at higher impact values." Please elaborate on why energy transfer decreases.

**Response:** Thanks for your valuable comment. We have added details on why the energy transfer decreases in the revised manuscript. "At higher impact values, the released energy was not fully used in the wave formation and propagation; instead, a part of the energy was dissipated in the formation of splash, and in the formation of air pockets and their subsequent collapse".

5. The conclusions presented in the manuscript as a list of discrete findings, although these individual findings are valuable and provide a deep insight into your results. But it is recommended to provide a clear, most critical implication of your work. For instance, granular slides underestimate the hazard caused by rotational cliff collapse and the effect of water depth on the induced water waves.

**Response:** Following the suggestion, we have provided the clear implications of our study in the conclusions: "Research findings highlight that accurate hazard assessment of the cliff collapse

requires models that account for the rotational failure mode and the fragmentation upon impact with the water surface. Traditional granular slide models may result in an underestimation of the initial wave amplitude and energy transferred".

6. To enhance the **quality and clarity** of the **figures** in the manuscript, it is strongly recommended to revise all the figures (preferably using Origin and/or MATLAB) and add more relevant explanation to the respective captions (they must include relevant details such as the data source, experimental conditions, and any important observations or trends depicted in the figure).

**Response:** Following the instructions, all the figures have been redrawn using Origin, and more relavent explaination has been added to the captions of following figures in the revised manuscript.

Fig. 6: (a) Dimensionless impact energy (K.Ε/ρgh³) vs relative wave amplitude, indicating a nonlinear trend, (b) Wave energy vs relative wave amplitude, indicating higher wave amplifications in shallow waters.

Fig. 7: Relative wave amplitude vs relative wave runup at various slope angles and water depth.

Fig. 10: Relative wave amplitude for various water depths, cliff height, and fall height at 30° runup slope angle, (a, b&c) relative wave amplitude induced by 0.245 m fall height, (d, e&f) relative wave amplitude induced by 0.445 m fall height, (g, h&i) relative wave amplitude induced by 0.645 m fall height.

Fig. 11: Water waves induced by equivalent granular mass at 30° slope angle.

7. Kindly **rewrite the abstract and conclusion** of the research article by providing a detailed description of the main results and the methodical steps used to achieve them. Highlight the novelty in the Abstract. Please include specific quantitative values

**Response:** According to your suggestion, the abstract has been modified and rewritten. Explanation can be seen in the first comment. Moreover, the conclusions have also been revised according to the instructions.

- It was concluded that water depth strongly controls the shape of the induced splash and wave amplification. Shallow water induced elongated, tall splashes, and higher wave amplitudes; in contrast, deep water produced mushroom-shaped splashes with higher energy dissipation and lower wave amplitudes.
- 2. The higher values of Froude number (> 1.2) for all the experiments indicate that the viscous effects were negligible, so the Froude number was selected as the most suitable dynamic

- scaling factor for describing the behaviour of the waves.
- 3. The wave amplitude was greatly influenced by cliff height (51 %), number of fragments (22 %), Impact velocity (18 %), cliff mass (4.69 %), and water depth (4.36 %). Whereas the wave runup was governed by the runup slope angle, impact velocity, and cliff mass.
- 4. The amplitude of the wave induced by equivalent granular mass sliding on a 30° slope was 28-42% lower than the waves induced by rotational cliff collapse, thus concluding that the mode of energy transfer to the water body plays a critical role in wave dynamics.
- 5. A novel MEP-based prediction model was developed for wave amplitude and runup. The model showed great performance during the training and testing stage, and showed high sensitivity to the used parameters, thus confirming its reliability.
- 6. Research findings highlight that accurate hazard assessment of the cliff collapse requires models that account for the rotational failure mode and the fragmentation upon impact with the water surface. Traditional granular slide models may result in an underestimation of the initial wave amplitude and energy transferred.
  - 8. In the **conclusion** section, kindly provide a comprehensive summary of the main findings of the study, including the novelty of the approach used and its potential applications in engineering?

Response: We are thankful to the reviewer for their valuable comment. The conclusions have been revised and can be found in the previous comment. Moreover, a paragraph explaining the novelty has already been added to the manuscript. Furthermore, the last conclusion in the revised manuscript highlights its potential application in engineering. "Research findings highlight that accurate hazard assessment of the cliff collapse requires models that account for the rotational failure mode and the fragmentation upon impact with the water surface. Traditional granular slide models may result in an underestimation of the initial wave amplitude and energy transferred."

9. Please enhance the **readability of the paper**. A concisely presented paper with high readability can improve the impact of the article. Addressing these aspects would significantly enhance the scientific rigor and practical applicability of the study.

**Response:** Following the instructions, the readability of the manuscript has been improved, and the abstract and the introduction have been revised. Sentence structure has been improved.

10. Please ensure the **referencing** is relevant, up to date, and accessible to our international readers. Please cite only references that are relevant and absolutely necessary. Papers with TOO MANY references are generally not acceptable. It is strongly recommended to declare your total self-citations if you haven't done so (max 5 or 20% of total references, whichever is smaller).

**Response:** Thanks for your valuable insights. We have tried to add the most relevant references to the manuscript, since the manuscript covers three aspects, i.e., experimental, numerical, and prediction modeling, which is why there are a bit more references. Nevertheless, we have removed marginally relevant and redundant citations from the text. The total number of self-citations remains less than 20% of the total citations and fewer than 5.

---

## Author Response (AR1)

**Response to the comments**

**1. Response to refree-1 comments**

This manuscript discusses the hydrodynamics of waves generated by rotational cliff collapses, with a specific and novel focus on the critical role of cliff fragmentation upon impact with the water surface. The study addresses a significant gap in the literature, as the disintegration of the sliding mass is a prevalent yet often oversimplified phenomenon in existing models. By systematically exploring the effects of cliff fragmentation upon impact with the water surface, on wave amplitude and runup, the authors provide valuable insights that are highly beneficial for improving hazard assessment and risk mitigation strategies in coastal environments. The topic is of considerable interest to the broader scientific community, particularly in the fields of geohazards, coastal engineering, and fluid dynamics. However, while the study's premise is compelling and its core contribution is novel, the manuscript, in its current form, requires major revision. The following observations need to be addressed before publication.

1. The manuscript requires thorough proofreading to improve clarity and readability. The language, with numerous grammatical errors and awkward phrasings, hinders the effective communication of the science. I recommend a comprehensive revision of the text by a native English speaker or a professional editing service.

**Response:** Following the advice, we have found a native speaker and have improved the overall sentence structure and grammar in the revised manuscript.

2. Line 14 should be replaced as the granular material/block is sliding down.

**Response:** The mentioned change has been incorporated into the revised manuscript. Lines (13-16)

3. Line 23-26, "A comparison between the wave induced by fragmented cliff collapse and an equivalent amount of granular mass sliding from a 30° slope indicates that the amplitude of the waves induced by granular mass is 42%, 35%, and 28% less than that of a fragmented cliff collapse." It is recommended to write in reverse order, i.e., the amplitude induced by rotational fall is more than the sliding.

**Response:** Following the advice, the changes have been made in the revised manuscript. "Moreover, compared to an equivalent amount of granular mass sliding down a 30° slope, rotational cliff collapse produced 28-42% higher wave amplitudes due to the acute impact that transfers energy more efficiently". Lines (19-22)

4. It is mentioned that the authors performed experimental and numerical modeling and then developed a prediction model; however, I couldn't find any information on the numerical modeling in the abstract. It would be beneficial to include some information on numerical modeling as well.

**Response:** According to the instructions by the reviewer, the information on the numerical modeling, "Results indicate that as the water depth decreased, the impact Froude number and relative wave amplitude increased, wave velocity decreased, and splash showed greater elongation. The numerical modelling results also confirmed the experimental trends." Lines (16-19)

5. Lines 101 to 106 reference a format error.

**Response:** All the references have been corrected in the revised manuscript.

6. Line 125 " Scientists have M. M. Das and Wiegel (1972) proposed…, doesn't make sense.

**Response:** We appreciate the reviewer's profundity. We have corrected it in the revised manuscript. Line-94

7. Line 223, the dimensions of the single block need to be checked. Your experimental flume is 0.5 m wide, and the single block length is 0.55 m. How?

**Response:** Thanks for pointing out such a grave typographical mistake; actual dimensions are 0.055x0.05x0.042 m. Corrections have been made in the revised manuscript. Lines (200-201)

8. The statement "the blocks were joined together with the help of cement paste having water-cement ratio W/C 0.8 and cured for 2 hours in front of an electric heater, such that the bond is weak enough that it fragments at the joints upon impacting the water surface." Needs to be backed up with reasonable arguments.

**Response:** The purpose of using a high water cement ratio and short curing duration was to deliberately create weak inter-block bonds that fragment upon impact with the water surface, thereby replicating the brittle joint failure that is observed in actual rotational cliff collapses. The real cliffs mass often consists of stratified material with preexisting fractures and low interlocking bonds. Therefore, the weak bonding was selected so that it fragments when it impacts the water surface. The short curing provided sufficient hardness for handling while maintaining low tensile bonds. We have added more details in the revised manuscript. "To ensure the weak bond strength, several trials for bond strength were carried out after a curing period of 2 hours, and it was found to be in the range of 0.42-0.5 MPa. In contrast, the inertial stresses at the time of impact were several times higher, such that they caused the fragmentation of the cliff. This condition was purposely designed to imitate naturally fractured cliff materials, confirming that the structure fragmented primarily along the joints upon impact with the water surface, consistent with field observations of rotational cliff collapses." Lines (205-212)

9. The sentence "To avoid the slippage of blocks and to replicate field conditions, fine-grained bricks of the same material as the cliff were pasted on the rotational platform" needs to be corrected.

**Response:** Following the advice, we have corrected it in the revised manuscript. " To avoid the slippage of blocks and to ensure that it had sufficient frictional resistance needed for pure rotational motion of the simulated cliff, finely-grounded bricks of the same cliff material were pasted on the rotational platform, thereby preventing translational motion or vertical free fall into the water." Lines (215-219)

10. The discussion of splash shape requires further clarification. In particular, the transition from an elongated splash at lower water depths to a mushroom-shaped splash at greater depths is described qualitatively but not sufficiently explained in terms of the underlying hydrodynamics. It is unclear whether this change is primarily governed by momentum dissipation, confinement effects due to water depth, or interactions between fragment number, impact velocity, and water depth. Could the authors elaborate on the physical mechanisms driving this transition, and indicate whether the observed shapes are consistent across multiple trials or strongly dependent on other control parameters?

**Response:** Reviewer has raised a valid point. Based on the experimental results, the elongated splash observed at shallow water depth arises from reduced vertical confinement of the impact momentum. At lower depth, the fragments' momentum penetrates rapidly to the bottom surface, limiting vertical jet development and instead elongating the splash outward along the surface. Consequently, at greater water depth, the momentum dissipates before interacting with the bottom surface, resulting in a vertical jet and the formation of a mushroom-shaped splash. This transition was observed across repeated trials and was primarily controlled by water depth relative to the fall height of the cliff fragments. Secondary parameters, such as the number of fragments and impact velocity, modulated the intensity of the splash and wave height. The shape of the splash, i.e., elongated or mushroom type, was consistently reproduced under the respective shallow and deep water depths. Thus, the observed behavior highlights water depth as the dominant factor in determining splash geometry in our study. Specific details have been added in the revised manuscript. Lines (332-333 & 338-339).

11. In section 2.3, you have stated that the VOF method is chosen. What are other numerical techniques that are used to simulate two-phase flows? It is understandable that for the current work, VOF might be better, but it would be important to mention those other methods briefly in this section as well, in order to have a complete picture of available numerical schemes.

**Response:** We thank the reviewer for this constructive feedback. The following paragraph has now been added to the manuscript in section 2.3 to give a holistic picture of the available numerical schemes."Alternative numerical schemes, such as the Front Tracking approach, are generally limited in handling complex topological changes (Tryggvason et al., 2001). Another approach is the Level Set method, but it suffers from mass conservation and convergence issues. The Lattice Boltzmann Method (LBM) is also common; however, its applicability to high velocity impact is rather limited (Aidun & Clausen, 2010)." Line (241-247).

Aidun, C. K., & Clausen, J. R. (2010). Lattice-Boltzmann method for complex flows. *Annual Review of Fluid Mechanics*, *42*(1), 439-472.

Tryggvason, G., Bunner, B., Esmaeeli, A., Juric, D., Al-Rawahi, N., Tauber, W.,Jan, Y.-J. (2001). A front-tracking method for the computations of multiphase flow. *Journal of computational physics*, *169*(2), 708-759.

12. What are the specific boundary conditions used in the simulation setup? Please mention it alongside the software used and numerical schemes, as these specific details help reproduce the work.

**Response:** We thank the reviewer for this valuable comment. To further strengthen clarity, we have now added the boundary conditions of the simulation setup in the manuscript. "The bottom boundary was modeled as a no-slip wall, while the top boundary was set as a pressure outlet at atmospheric conditions, and the lateral sides were modeled as stationary walls to confine the liquid film within the domain." The details on the boundary conditions have been incorporated in the revised manuscript. Lines (271-274)

13. Most importantly, the accuracy of water wave amplitude and runup prediction is highly sensitive to the selection of hyperparameters (such as population size, number of generations, and mutation/crossover rates). Inadequate tuning may lead to premature convergence, underfitting, or unnecessarily high computational cost. How did the authors consider this aspect?

**Response:** We have addressed the concern about hyperparameter sensitivity in Multi-Expression Programming (MEP). During model development, prerequisite tuning procedures were applied to optimize key hyperparameters, including population size, number of generations, and mutation/crossover rates. This careful selection minimized the risk of premature convergence or underfitting while ensuring computational efficiency. The details have been added in the revised manuscript. Lines (326-328).

14. Please explain that while a high $R^2$ indicates strong correlation between predicted and observed values, relying solely on it may give a misleading impression of model quality. For wave prediction, capturing extreme or rare events is critical, and $R^2$ does not fully reflect this capability.

**Response:** The observation about the limitations of relying solely on $R^2$ has also been taken into consideration. While $R^2$ was employed as a comparative performance indicator, additional emphasis was placed on the developed MEP model's ability to capture variability in both typical and extreme wave conditions. This ensured that the evaluation framework not only

relied on statistical correlation but also reflected the robustness and practical reliability of predictions in diverse scenarios.

**2. Response to referee-2 comments**

We are thankful to Reviewer (RC2) for their time and valuable feedback on our manuscript. We have carefully addressed and justified all the observations made in the review. Text in red color highlights the changes made in the revised manuscript.

1. Please ensure the **abstract** is short but reflects the approach, results, and conclusions correctly and concisely. Please check the keywords and highlights to ensure they are appropriate and complete. Highlights should be very brief and to the point and attractive to the readers of this journal. Kindly rearrange the keywords according to alphabetical order.

**Response:** Following the suggestion, the abstract has been modified, and keywords have been arranged according to alphabetical order. "Cliff collapses in small lakes, and reservoirs induce powerful waves, threatening the offshore infrastructure. Unlike previous studies on waves induced by granular slide, this study experimentally and numerically investigates the waves induced by rotational cliff collapse, whereby the cliff fragments upon impact with the water surface, and determines the wave amplitude, runup, and energy transfer mechanics. Results indicate that as the water depth decreased, the impact Froude number and relative wave amplitude increased, wave velocity decreased, and splash showed greater elongation. The numerical modelling results also confirmed the experimental trends. Moreover, compared to an equivalent amount of granular mass sliding down a 30° slope, rotational cliff collapse produced 28-42% higher wave amplitudes due to the acute impact that transfers energy more efficiently. Machine learning based prediction models were subsequently developed to predict the wave amplitude and runup. The prediction models performed well both in the training and testing stages, with high $R^2$ values, and were validated via established statistical indices, sensitivity, and parametric analysis. The prediction models highlighted a cumulative 90% contribution of impact velocity, cliff height, and the number of fragments on the wave amplitude. In comparison, runup is greatly influenced by bank slope angle, impact velocity, cliff mass, and height. The experimental results and developed prediction models can provide the basis for understanding the rotational cliff collapse-induced waves and can help with disaster mitigation and risk assessment by effectively predicting the wave amplitude and

runup.

Keywords: Cliff fragmentation; landslide tsunami; prediction models; rotational cliff collapse; wave amplitude, and runup." Lines (12-33)

2. The paper lacks a comprehensive background and literature review section in a **tabulated form**. To enhance the significance of the study, the authors should include an in-depth review of related literature. To enhance the clarity and comprehensiveness of the introduction section, the authors are kindly requested recommended to include a pertinent   table.

**Response:** We thank the reviewer for the suggestion. A comprehensive literature summary on the existing prediction models for the water wave amplitude and runup is already included in the original manuscript as Table 1. This table does not duplicate the literature that has been discussed in the text; rather, it provides a detailed summary of the key studies related to prediction modelling of water waves, thus highlighting the most relevant and fundamental work in the field of wave mechanics.

| Authors | Predictive model |
|---|---|
| (Kamphuis and Bowering, 1970) | $A_m = \left(\dfrac{v_s}{\sqrt{gh}}\right)^{0.7}\left(0.31 + 0.2\,log\left(\dfrac{l_s}{h^2}\right)\right) + 0.35e^{-0.08(x/h)}$ |
| (Noda, 1970) | $A_m = 1.32\,\dfrac{v_s}{\sqrt{gh}}$ |
| (Huber and Hager, 1997) | $\dfrac{H_m}{h} = 2 \times 0.88\,sin\,\theta\,cos^2\left(\dfrac{2\alpha}{3}\right)\left(\dfrac{\rho_5}{\rho_w}\right)^{0.25}\left(\dfrac{V}{wh^2}\right)^{0.5}\left(\dfrac{r}{h}\right)^{-\frac{2}{3}}$ |
| (Fritz et al., 2004) | $A_m = 0.25\left(\dfrac{v_s}{\sqrt{g_h}}\right)^{1.4}\left(\dfrac{S}{h}\right)^{0.8}$ |
| (Panizzo et al., 2005) | $\dfrac{H_m}{h} = 0.07\left(\dfrac{T_s h^2}{ws}\right)^{-0.45}(sin\,\alpha)^{-0.88}e^{0.6cos\theta}\left(\dfrac{r}{n}\right)^{-0.44}$ |
| (Heller, 2007) | $A_m = \dfrac{4}{9}\left[F\left(\dfrac{S}{h}\right)^{1/2}\rho^{1/4}\left(cos\,\dfrac{6\alpha}{7}\right)^2\right]^{4/5}$ |
| (Mohammed and Fritz, 2012b) | $A_m = max(A_{C_1}, A_{C2})$
 $A_{c1} = 0.3F^{2.1}\left(\dfrac{s}{h}\right)^{0.6}\left(\dfrac{r}{h}\right)^{\left(-1.2F^{0.25}\left(\frac{s}{h}\right)^{-0.02}w-0.33/h\right)}cos\alpha$
 $A_{c2} = 1.0FS^{0.8}\left(\dfrac{w}{h}\right)^{-0.4}\left(\dfrac{l}{h}\right)^{-0.5}\left(\dfrac{\gamma}{h}\right)^{-1.5F^{0.5}\left(\frac{w}{h}\right)^{-0.07}\left(\frac{w}{h}\right)^{-0.3}}cos^2\,\alpha$ |
| (Wang et al., 2016b) | $A_m = 1.17F\left(\dfrac{sl}{bh}\right)^{0.25}\left(\dfrac{w}{b}\right)^{0.45}(Sin^2\alpha + 0.6cos^2\alpha)$ |
| (Li et al., 2023b) | $A_m = 0.59\sqrt{\dfrac{2H(1-fcot\alpha)}{h}}\left(\dfrac{swl}{h^3}\right)^{N-0.11}\left(\dfrac{x}{h}\right)^{-0.43}cos^2\left(\dfrac{2}{3}\alpha\right)$ |

*Note*: $l$ is the landslide length; $s$ is the landslide thickness; $w$ is the landslide width; m is the landslide mass weight; $V$ is the landslide volume; $H$ is the landslide height; $T_s$ time for motion of slide, $b$ is the river width; $h$ is the still water depth; $x(r)$ is the offshore distance from the bank slope; $\alpha$ is the slope angle; $\theta$ is the angular direction; $v_s$ is the impact velocity.

3.  I am curious about the 'weak bond' as it is a critical parameter. It is recommended to provide a quantitative measure of bond strength. For example, what strength did you achieve by using 0.8 w/c and curing it for two hours? And how do you compare it with the inertial stresses upon impact with the water surface? This would allow others to reproduce the phenomenon and get results."

**Response:** We acknowledge the reviewer's observation, we have provided a detail on the bond strength in the revised manuscript. "To ensure the weak bond strength, several trials for bond strength were carried out after a curing period of 2 hours, and it was found to be in the range of 0.42-0.5 MPa. In contrast, the inertial stresses at the time of impact were several times higher, such that they caused the fragmentation of the cliff. This condition was purposely designed to imitate naturally fractured cliff materials, confirming that the structure fragmented primarily along the joints upon impact with the water surface, consistent with field observations of rotational cliff collapses". Lines (205-212)

4.  "The negative quadratic coefficient in Fig. 6(a) indicates a nonlinear response, such that at the start the wave amplitude increases as the impact energy increases, but after a certain value it decreases, due to reduced energy transfer at higher impact values." Please elaborate on why energy transfer decreases.

**Response:** Thanks for your valuable comment. We have added details on why the energy transfer decreases in the revised manuscript. "At higher impact values, the released energy was not fully used in the wave formation and propagation; instead, a part of the energy was dissipated in the formation of splash, and in the formation of air pockets and their subsequent collapse". Lines (361-364).

5.  The conclusions presented in the manuscript as a list of discrete findings, although these individual findings are valuable and provide a deep insight into your results. But it is recommended to provide a clear, most critical implication of your work. For instance, granular slides underestimate the hazard caused by rotational cliff collapse and the effect of water depth on the induced water waves.

**Response:** Following the suggestion, we have provided the clear implications of our study in the

conclusions: "Research findings highlight that accurate hazard assessment of the cliff collapse requires models that account for the rotational failure mode and the fragmentation upon impact with the water surface. Traditional granular slide models may result in an underestimation of the initial wave amplitude and energy transferred". Lines (780-783)

6. To enhance the **quality and clarity** of the **figures** in the manuscript, it is strongly recommended to revise all the figures (preferably using Origin and/or MATLAB) and add more relevant explanations to the respective captions (they must include relevant details such as the data source, experimental conditions, and any important observations or trends depicted in the figure).

**Response:** Following the instructions, all the figures have been redrawn using Origin, and a more relevant explanation has been added to the captions of the following figures in the revised manuscript.

[Figure]

Fig. 6: (a) Dimensionless impact energy (K.E/$\rho gh^3$) vs relative wave amplitude, indicating a nonlinear trend, (b) Wave energy vs relative wave amplitude, indicating higher wave amplifications in shallow waters.

[Figure]

Fig. 7: Relative wave amplitude vs relative wave runup at various slope angles and water depth.

[Figure]

Fig. 10: Relative wave amplitude for various water depths, cliff height, and fall height at 30°runup slope angle, (a, b&c) relative wave amplitude induced by 0.245 m fall height, (d, e&f) relative wave amplitude induced by 0.445 m fall height, (g, h&i) relative wave amplitude induced by 0.645 m fall height.

[Figure]

Fig. 11: Water waves induced by equivalent granular mass at 30° slope angle。

7.  Kindly **rewrite the abstract and conclusion** of the research article by providing a detailed description of the main results and the methodical steps used to achieve them. Highlight the novelty in the Abstract.   Please include specific quantitative values

**Response:** According to your suggestion, the abstract has been modified and rewritten. Explanation can be seen in the first comment,  Lines (12-33) of the revised manuscript. Moreover, the conclusions have also been revised according to the instructions. Lines (762-783)

1.  It was concluded that water depth strongly controls the shape of the induced splash and wave amplification. Shallow water induced elongated, tall splashes, and higher wave amplitudes; in contrast, deep water produced mushroom-shaped splashes with higher energy dissipation and lower wave amplitudes.

2.  The higher values of Froude number (> 1.2) for all the experiments indicate that the viscous

effects were negligible, so the Froude number was selected as the most suitable dynamic scaling factor for describing the behaviour of the waves.

3. The wave amplitude was greatly influenced by cliff height (51 %), number of fragments (22 %), Impact velocity (18 %), cliff mass (4.69 %), and water depth (4.36 %). Whereas the wave runup was governed by the runup slope angle, impact velocity, and cliff mass.

4. The amplitude of the wave induced by equivalent granular mass sliding on a 30° slope was 28-42% lower than the waves induced by rotational cliff collapse, thus concluding that the mode of energy transfer to the water body plays a critical role in wave dynamics.

5. A novel MEP-based prediction model was developed for wave amplitude and runup. The model showed great performance during the training and testing stage, and showed high sensitivity to the used parameters, thus confirming its reliability.

6. Research findings highlight that accurate hazard assessment of the cliff collapse requires models that account for the rotational failure mode and the fragmentation upon impact with the water surface. Traditional granular slide models may result in an underestimation of the initial wave amplitude and energy transferred.

8. In the **conclusion** section, kindly provide a comprehensive summary of the main findings of the study, including the novelty of the approach used and its potential applications in engineering?

**Response:** We are thankful to the reviewer for their valuable comment. The conclusions have been revised and can be found in the previous comment. Moreover, a paragraph explaining the novelty has already been added to the manuscript. Furthermore, the last conclusion in the revised manuscript highlights its potential application in engineering. "Research findings highlight that accurate hazard assessment of the cliff collapse requires models that account for the rotational failure mode and the fragmentation upon impact with the water surface. Traditional granular slide models may result in an underestimation of the initial wave amplitude and energy transferred." Lines (780-783).

9. Please enhance the **readability of the paper**. A concisely presented paper with high readability can improve the impact of the article. Addressing these aspects would significantly enhance the scientific rigor and practical applicability of the study.

**Response:** Following the instructions, the readability of the manuscript has been improved, and the abstract and the introduction have been revised. Sentence structure has been improved.

10. Please ensure the **referencing** is relevant, up to date, and accessible to our international readers. Please cite only references that are relevant and absolutely necessary. Papers with TOO MANY references are generally not acceptable. It is strongly recommended to declare your total self-citations if you haven't done so (max 5 or 20% of total references, whichever is smaller).

**Response:** Thanks for your valuable insights. We have tried to add the most relevant references to the manuscript, since the manuscript covers three aspects, i.e., experimental, numerical, and prediction modeling, which is why there are a bit more references. Nevertheless, we have removed marginally relevant and redundant citations from the text. The total number of self-citations remains less than 20% of the total citations and fewer than 5.